# Representation Surgery in Model Merging with Probabilistic Modeling

**Qi Wei**[1]   **Shuo He**[1]   **Enneng Yang**[1]   **Tingcong Liu**[1]   **Haobo Wang**[2]   **Lei Feng**[3]   **Bo An**[1 4]

## Abstract

Model merging aims to achieve multitask performance by merging multiple expert models without the need to access the raw training data. Recent research identified the *representation bias* of model merging, characterized by a discrepancy in the representation distribution between the merged and individual models, hindering the performance of model merging methods. To mitigate this bias, a task-specific MLP, Surgery, was built to model the bias that is subsequently decreased on the merged representation. However, this strategy is still suboptimal due to the limited modeling capability within the deterministic manner. To address this issue, we present ProbSurgery, a probabilistic module specifically designed to accurately model the representation bias. This module generates an embedding distribution for each sample and outputs the representation bias through a sampling process. ProbSurgery offers superior representational capacity by naturally handling the uncertainty resulting from parameter interference of merging multiple models. Besides, we provide a theoretical analysis to reveal the advance of the probabilistic manner and propose an extension of ProSurgery for adapting to the task-sharing setting. Extensive experiments verify the effectiveness of ProbSurgery while maintaining generalization capabilities in real-world scenarios. The code is now available at this url.

## 1. Introduction

Multi-task learning (MTL) (Caruana, 1997) is a machine learning approach that simultaneously learns multiple related tasks to improve model generalization and efficiency. Compared with traditional single-task learning that focuses solely on a specific task, MTL leverages the shared knowledge and representations among tasks, enhancing the overall performance (He & Lawrence, 2011). However, constrained by a rigid training paradigm, i.e., "data collection first and then jointly training", MTL raises several concerns: 1) the high cost of data collection as well as the risk of privacy leakage, and 2) limited flexibility, as introducing a new task necessitates retraining the MTL model on a combination of both old and new (tasks') datasets.

Recently, model merging has attracted increasing attention (Matena & Raffel, 2022; Ilharco et al., 2023; Yadav et al., 2023; Yu et al., 2024; Yang et al., 2024c;a), as it offers a new approach to fulfilling MTL without requiring training data. This approach is encouraged by the growth of open-source communities like `Huggingface`[1] and recognizes that published trained (or fine-tuned) models can be treated as resources, similar to datasets. Therefore, a model can obtain multi-task knowledge by adapting the parameters of multiple existing expert models that share a consistent network structure instead of training from scratch on a task-combined dataset. In the era of foundation models, the advantages of *data independence* and *training free* make model merging more practical.

However, models merged through existing methods (Matena & Raffel, 2022; Ilharco et al., 2023; Yadav et al., 2023; Yang et al., 2024c) often experience performance degradation, making them hardly comparable to traditional MTL methods (Sun et al., 2020b; Royer et al., 2024), let alone that of individual models. A recent study (Yang et al., 2024a) analyzed this issue from a representation perspective and discovered a discrepancy in the representation distribution between merged and individual models, termed *representation bias*, leading to degenerated performance. To mitigate this bias, the authors introduced a task-specific, learnable module after the encoder, termed Surgery, to refine the biased representation. With the Surgery module, the merged model attains representational capabilities close to those of individually trained models.

Despite notable improvements achieved, we argue that there are still two concerns about the Surgery module:

[1]Nanyang Technological University, Singapore [2]Zhejiang University, China [3]Southeast University, China [4]Skywork AI, Singapore. Correspondence to: Lei Feng <fenglei@seu.edu.cn>.

*Proceedings of the $42^{nd}$ International Conference on Machine Learning*, Vancouver, Canada. PMLR 267, 2025. Copyright 2025 by the author(s).

[1]`https://huggingface.co`. For example, HuggingFace hosts a collection of over 1.7 million pretrained models.

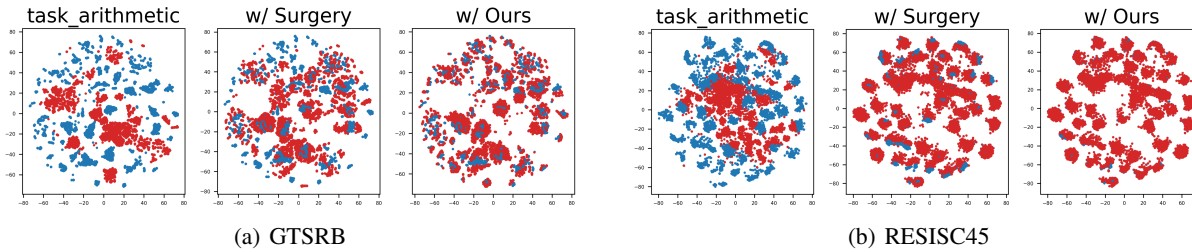

Figure 1. Comparisons of varying generated representations, given *Task Arithmetic* as the baseline and ViT-B/32 as the backbone. We can see that the feature rectified by Surgery (the red points) is not totally aligned with the optimal (the blue points).

- *(Insufficient performance) Has the representation discrepancy been fully addressed?* We visualize the representations learned with Surgery across different methods in Figure 1. The results indicate that the rectified representations via Surgery still remain considerably distant from the optimal ones, which cannot be correctly tackled by the subsequent classification head.

- *(Limited applicability) Can multiple task-specific Surgery modules be substituted with a single task-shared one?* The Surgery module is task-specific which relies on prior knowledge about the task ID during inference. Ideally, we aim to construct a single but powerful module, akin to a meta-model, capable of handling biased representations across all tasks. It provides fewer parameters and better applicability in real-world applications.

To address these concerns, we deeply dig into modeling the representation bias by the Surgery mechanism and discover its probabilistic understanding in this paper. Specifically, rather than the deterministic manner employed in Surgery, which is constricted by limited representation capability, we reformulate the representation bias as a distribution whose parameters are determined by MLPs in an amortized fashion. We refer to this module as *ProbSurgery* in our work. Then, the representation bias can be obtained by sampling from the generated distribution. In contrast to Surgery, ProbSurgery presents two advantages, including 1) better representation capability: Due to significant interference among the parameters of multiple models when merging, the generated representation exhibits uncertainty. Probabilistic representation learning can effectively model the uncertainty. 2) Superior applicability: One single ProbSurgery can be directly utilized for tackling all merging tasks (named one-to-all setting in this paper), whose performance even outperforms that of Surgery trained by the task-specific setting. This probabilistic reformulation not only improves the flexibility and robustness of the model but also broadens its applicability to diverse and challenging real-world scenarios. Eventually, we further provide a theoretical analysis for ProbSurgery based on a *PAC-Bayes framework* (McAllester, 2003), uncovering that modeling representation bias as a distribution yields lower classification error.

Our contributions are summarized as threefold:

- We propose a probabilistic representation surgery module, named ProbSurgery, as post-calibration to correct biased representations resulting from model merging, which can consistently enhance the performance of existing model merging methods.

- We provide theoretical analysis for ProbSurgery, which shows that modeling representation bias as a distribution yields lower classification error.

- We propose two strategies for extending ProbSurgery to the one-to-all setting, i.e., learning one ProbSurgery module for all merged tasks, which offers practical values.

Advancing performance and extensive analyses demonstrate the effectiveness and applicability of ProbSurgery.

## 2. Related Work

### 2.1. Model Merging for Multi-Task Learning

Referring to Yang et al. (2024a), we divide existing model merging methods into three types based on the stage they focus on: *before*, *during*, and *after* merging.

1) The methods that operate *before merging* mainly focus on providing valuable prior knowledge for subsequent model merging. For example, Ortiz-Jimenez et al. (2024) add a constraint when fine-tuning the pretrained model in different tasks, i.e., updating the model's parameters in independent Tangent spaces (Jacot et al., 2018). Similarly, one recent work, DARE (Yu et al., 2024), proposes to force the offsets of randomly selected parameters (above 90%) to remain unchanged. Thus, different tasks can benefit from an individual parameter subspace and reduce performance losses.

2) The majority of works in model merging pay attention to mitigate the issue of interference and conflicts with different models, which belong to a family of *during merging*. Previous studies such as Weighted Averaging (Li et al., 2023), Ties-Merging (Yadav et al., 2023), and AdaMerging (Yang et al., 2024c) typically pre-set or learn task-specific weights to adjust the contributions of different tasks to the final merged model. In the past year, a lot of methods (Lu et al., 2024; Tang et al., 2024; Huang et al., 2024) based on Mixture-of-experts or Router technique (Masoudnia &

Ebrahimpour, 2014) were proposed to split shared and task-specific knowledge and then dynamically select them for tackling downstream tasks after merging.

3) The methods that operate *after merging* (Yang et al., 2024a;b) are proposed to rectify the biased representations generated after merging. The most representative method, Surgery (Yang et al., 2024a), constructs a lightweight, task-specific module to capture the bias between the optimal and interfered representations. A key merit of these methods lies in their seamless integration with weight-space merging methods, thereby enhancing the overall performance.

In this paper, we design a post-calibration approach derived from the Surgery framework in a probabilistic manner, offering enhanced generalization and representation capabilities. ProbSurgery can be integrated into existing merging methods and further boost their performance.

## 2.2. Probabilistic Representations

Learning representations in a stochastic embedding space first emerged in word embeddings (Vilnis & McCallum, 2014) and has since been widely applied to various natural language processing tasks (Li et al., 2018; Neelakantan et al., 2015). Meanwhile, the probabilistic representations are also adapted into various vision tasks like face recognition (Chang et al., 2020; Shi & Jain, 2019), human pose estimation (Sun et al., 2020a), prototype-based few-shot learning (Scott et al., 2019) and video understanding (Park et al., 2022). Existing methods demonstrate that probabilistic representations contribute to uncertainty modeling and improving the model's generalization (Li et al., 2018).

## 3. Preliminaries

**Problem Definitions.** Consider a set of models denoted as $\{f_{\boldsymbol{\theta}_1}, \ldots, f_{\boldsymbol{\theta}_T}\}$, fine-tuned from a base model $f_{\boldsymbol{\theta}_{\mathrm{base}}}$ on $T$ individual tasks. In the context of model merging, our objective is to integrate these parameters $\{\boldsymbol{\theta}_t\}_{t=1}^T$ into a unified one $\boldsymbol{\theta}_{\mathrm{unif}}$, where the resulting model, $f_{\boldsymbol{\theta}_{\mathrm{unif}}}$, can achieve consistently strong generalization across all $T$ tasks.

Formally, we aim to obtain a merged model $f_{\boldsymbol{\theta}_{\mathrm{unif}}}$ that can fulfill the minimal loss value on the test dataset $\{D_t^{te}\}_{t=1}^T$ of all tasks, which can be formulated by this objective $\min \frac{1}{T} \sum_{t=1}^T \mathbb{E}_{(\boldsymbol{x},y) \sim D_t^{te}} \ell(f_{\boldsymbol{\theta}_{\mathrm{unif}}}(\boldsymbol{x}), y)$, where $\ell$ denotes the loss function, such as the cross-entropy loss.

### 3.1. Representative Model Merging Methods

In this section, we first define some notations within model merging and introduce four representative merging methods.

**Definition 3.1.** *(Task vector). Assume a model $f_{\boldsymbol{\theta}_t}$ obtained by fine-tuning the base model $f_{\boldsymbol{\theta}_{\mathrm{base}}}$ on $t$-th task. The task vector $\boldsymbol{\tau}_t$ is defined as the element-wise difference between*

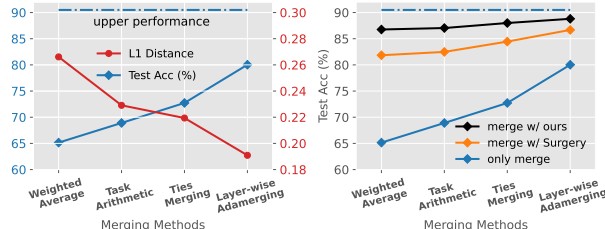

*Figure 2.* VIT-B/32 and 8 vision tasks. **Left:** Comparison of average performance and L1 representation distance regarding four methods on all vision tasks. **Right:** Performance improves after rectifying representations using Surgery or ours. *Upper performance* denotes the average performance of various expert models on corresponding merged tasks.

$\boldsymbol{\theta}_{\mathrm{base}}$ *and $\boldsymbol{\theta}_t$, i.e., $\boldsymbol{\tau}_t = \boldsymbol{\theta}_t - \boldsymbol{\theta}_{\mathrm{base}}$.*

Task vector can be regarded as the task-specific knowledge and has been applied for various scenarios, such as *knowledge transfer and adaptation*, *modeling task relationships*, and *task fusion in model merging* (Ilharco et al., 2023).

Based on these notations, we give the formal introduction to four model merging methods. ❶ *Weighted Average*, which directly averages the weight of models on $T$ tasks, is represented by $\boldsymbol{\theta}_{\mathrm{unif}} = \frac{1}{T} \sum_{t=1}^T \boldsymbol{\theta}_t$. ❷ *Task Arithmetic* (Ilharco et al., 2023), which merges the total $T$ task vectors into the base models' parameters, denoted by $\boldsymbol{\theta}_{\mathrm{unif}} = \boldsymbol{\theta}_{\mathrm{base}} + \lambda \sum_{t=1}^T \boldsymbol{\tau}_t$, where $\lambda$ is a hyper-parameter. ❸ *Ties-Merging* (Yadav et al., 2023), which splits the merging processes into three steps: TRIM, ELECT SIGN, and MERGE, where the first two steps aim to eliminate the symbol conflict problem in task vectors, denoted by $\phi(\cdot)$ for clarity. The merged weight can be written as $\boldsymbol{\theta}_{\mathrm{unif}} = \boldsymbol{\theta}_{\mathrm{base}} + \lambda \sum_{t=1}^T \phi(\boldsymbol{\tau}_t)$. ❹ *AdaMerging* (Yang et al., 2024c), which learns a set of task-wise or layer-wise coefficients for Task Arithmetic or Ties-Merging, denoted by $\boldsymbol{\theta}_{\mathrm{unif}} = \boldsymbol{\theta}_{\mathrm{base}} + \sum_{t=1}^T \lambda_t \boldsymbol{\tau}_t$ and $\boldsymbol{\theta}_{\mathrm{unif}} = \{\boldsymbol{\theta}_{\mathrm{base}}^l + \sum_{t=1}^T \lambda_t^l \boldsymbol{\tau}_t^l\}_{l=1}^L$ ($L$ denotes the total number of layers / blocks in the model $f$), respectively.

### 3.2. Representation Bias in Model Merging

Here, we first provide a formal definition of representation bias (Yang et al., 2024a) and then explain how it impacts the performance of merged models.

**Definition 3.2.** *(Representation bias). Consider $T$ expert models and the merged model, the representation bias on $t$-th task is defined as $\xi_t = \frac{1}{|D_t^{te}|} \sum_{i=1}^{|D_t^{te}|} \psi(z_i^{\boldsymbol{\theta}_{\mathrm{unif}}}, z_i^{\boldsymbol{\theta}_t})$, where $z$ denotes the extracted feature representation and $\psi(\cdot)$ is the arbitrary distance measure function.*

**Observation 3.3.** *Consider $T$ test datasets and individual expert models $\{D_t^{te}, f_{\boldsymbol{\theta}_t}\}_{t=1}^T$, the overall performance gap between the merged model $f_{\boldsymbol{\theta}_{\mathrm{unif}}}$ and these expert models obeys $G(f_{\boldsymbol{\theta}_{\mathrm{unif}}}, \{f_{\boldsymbol{\theta}_t}\}_{t=1}^T) = \mathbb{E}_{t \sim [T]}[Acc(f_{\boldsymbol{\theta}_t}, D_t^{te}) -$*

$Acc(\boldsymbol{f}_{\boldsymbol{\theta}_{\text{unif}}}, D_t^{te})] \propto \mathbb{E}_{t \sim [T]} \xi_t.$

This observation can be empirically validated (see Figure 2), which indicates that model merging methods that *minimize the gap between expert models and the merged model* are likely to achieve superior performance. Consequently, representation surgery emerges as a promising approach for further performance enhancement for the merged model.

## 4. Methodology

We divided our proposed methodology into two core parts:

- A probabilistic representation surgery module, named ProbSurgery, which enables more accurate modeling for representation bias and thus achieves better performance.

- A theoretical analysis for ProbSurgery and two strategies for extending ProbSurgery to the one-to-all setting, i.e., learning a single ProbSurgery module for all merged tasks.

ProbSurgery not only outperforms its counterpart, Surgery, as a post-calibration technique for model merging, the primary goal of our work, but also showcases markedly superior performance in tackling out-of-distribution and domain-shift challenges. Figure 3 shows the overall framework.

### 4.1. ProbSurgery

Previously, a post-merging representation surgery technique was proposed by Yang et al. (2024a), which aims to align the merged feature $\boldsymbol{z}^{\boldsymbol{\theta}_{\text{unif}}}$ with the individual model-generated feature $\boldsymbol{z}^{\boldsymbol{\theta}_t}$ for the task $t$ via a surgery module denoted by $\boldsymbol{g}_{\boldsymbol{\omega}}(\cdot)$, where the network $\boldsymbol{g}$ is modeled as a three-layer fully connected MLP and $\boldsymbol{\omega}$ denotes the learnable parameters (see Figure 3 (b)). The training objective of Surgery is:

$$\underset{\{\boldsymbol{\omega}_1,...,\boldsymbol{\omega}_T\}}{\arg\min} \mathcal{L}_{\text{align}} = \underset{\{\boldsymbol{\omega}_1,...,\boldsymbol{\omega}_T\}}{\arg\min} \sum_{t=1}^{T} \sum_{i=1}^{|D_t^{te}|} \psi\big(\boldsymbol{z}_{i,t}^{\boldsymbol{\theta}_{\text{unif}}} - \xi_{i,t}, \boldsymbol{z}_i^{\boldsymbol{\theta}_t}\big),$$
$$\text{where} \quad \xi_{i,t} = \boldsymbol{g}_{\boldsymbol{\omega}_t}\big(\boldsymbol{z}_{i,t}^{\boldsymbol{\theta}_{\text{unif}}}\big). \quad (1)$$

Totally, there are $T$ trainable modules $\{\boldsymbol{g}_{\boldsymbol{\omega}_1}, ..., \boldsymbol{g}_{\boldsymbol{\omega}_T}\}$ that are independently trained on $T$ task validation sets. Due to limited representation capability, one task-specific Surgery module can hardly be extended to other tasks.

To solve this issue, we propose a simple method that adapts the Surgery module in a probabilistic manner (see Figure 3). Firstly, we consider the same setting with Surgery (Yang et al., 2024a), i.e., a one-to-one setting that learns $T$ task-specific modules for $T$ individual tasks. Specifically, for a specific task[2], we formulate a probability distribution $p(\xi|Q)$ to represent the latent space $Q$ of the representation bias $\xi$, which is denoted by a Gaussian distribution:

$$p(\xi|Q) \sim \mathcal{N}(\mu, \sigma^2), \quad (2)$$

---
[2]For clarity, we omit the symbol of the task index $t$.

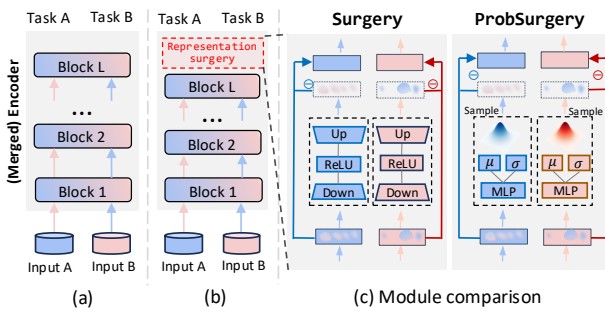

*Figure 3.* **Comparisons of different MTL frameworks.** (**a**) Typical model merging methods, which utilize the parameter fusion technique to fulfill one unified model that can handle multiple tasks simultaneously. (**b**) Post-merging representation surgery framework. The (last-layer) representation surgery module can be integrated with current model merging methods and boost their performance. (**c**) We compare our proposal, ProbSurgery, with Surgery (Yang et al., 2024a), which provides a probabilistic representation learning strategy for modeling the representation bias.

where $\mu$ and $\sigma$ are learned by ProSurgery, a lightweight, three-layer, fully connected MLP. From this representation distribution $p(\xi|Q)$, we sample an embedding $\hat{\xi}$ to represent the latent representation bias of the original one $\xi$, where a reparameterization trick (Kingma et al., 2015) is used as:

$$\hat{\xi} = \mu + \epsilon \cdot \sigma \quad \text{with} \quad \epsilon \sim \mathcal{N}(0, 1), \quad (3)$$

Correcting the biased embedding $\boldsymbol{z}^{\boldsymbol{\theta}_{\text{unif}}}$ by decreasing the probabilistic bias $\hat{\xi}$ offers the main advantage: the probabilistic approach inherently incorporates uncertainty to effectively model the biased embedding. This significantly enhances the capacity to resolve ambiguities arising from the merging of multiple models in parameter space.

Overall, the learning objective in our framework is

$$\underset{\{\boldsymbol{\omega}_1,...,\boldsymbol{\omega}_T\}}{\arg\min} \mathcal{L}_{\text{align}} + \lambda \mathcal{L}_{\text{reg}}$$
$$= \underset{\{\boldsymbol{\omega}_1,...,\boldsymbol{\omega}_T\}}{\arg\min} \sum_{t=1}^{T} \sum_{i=1}^{|D_t^{te}|} \psi\big(\boldsymbol{z}_{i,t}^{\boldsymbol{\theta}_{\text{unif}}} - \hat{\xi}_{i,t}, \boldsymbol{z}_i^{\boldsymbol{\theta}_t}\big) + \lambda L_{\text{KL}}^t, \quad (4)$$
$$\hat{\xi}_{i,t} = \phi(V_{\boldsymbol{\omega}_t}(\boldsymbol{z}_{i,t}^{\boldsymbol{\theta}_{\text{unif}}})) \quad \text{and} \quad L_{\text{KL}}^t = \text{KL}\big(p(\xi|Q) \| \mathcal{N}(0, 1)\big).$$

Note that $\lambda$ is a hyperparameter set to $1 \times 10^{-4}$, $\phi(\cdot)$ denotes the reparameterization trick, $V_{\boldsymbol{\omega}_t}$ denotes our ProSurgery module with learnable parameters $\boldsymbol{\omega}_t$, and $L_{\text{KL}}$ denotes the Kullback–Leibler (KL) divergence.

### 4.2. Theoretical Evidence

We theoretically demonstrate that, under a PAC-Bayes framework (McAllester, 2003), modeling representation bias as a distribution yields lower classification error.

Let $h(\cdot)$ denotes the fixed classifier (after the encoder). To correct the encoder's output, we introduce a bias vector

*Table 1.* Comparisons of different methods from four perspectives. Note that a larger number of "◯" denotes better performance. Combining ProbSurgery with DFA/PDA achieves a trade-off between parameter efficiency and performance.

| Methods | No Task ID | Parameter Efficient | Uncertainty Modeling | Performance | |
|---|---|---|---|---|---|
| | | | | Acc | Robustness |
| **Surgery** | ✗ | ✗ | ✗ | ◯ | ◯ |
| **ProbSurgery** | ✓ | ✗ | ✓ | ◯◯◯ | ◯◯ |
| w/ DFA | ✓ | ✓ | ✓ | ◯◯ | ◯◯◯ |
| w/ PDA | ✓ | ✓ | ✓ | ◯◯ | ◯◯◯ |

$\xi \in \mathbb{R}^d$ in subtractive form: the input to the classifier is $z - \xi$. We compare 1) the *deterministic bias*, a single vector $\xi_{\text{det}}$, and 2) the *probabilistic bias*, a distribution $Q_\xi$ from which we sample for each input. Our goal is to show that our probabilistic approach can yield a strictly smaller classification error bound than the deterministic one.

Firstly, we formally give the expected 0-1 classification error for our probabilistic method, as follows:

$$R_{\text{prob}} = \mathbb{E}_{(x,y)\sim\mathcal{D}^N} \mathbb{E}_{\hat{\xi}\sim Q(\hat{\xi}|z)}\big[\mathbb{1}(h(z - \hat{\xi}) \neq y)\big], \quad (5)$$

where $\mathbb{1}(\cdot)$ is an indicator function and $\mathcal{D}^N$ denotes the whole sample set of all merged tasks with a size of $N$.

**Theorem 4.1.** *(PAC-Bayes Theorem, adapted from (McAllester, 2003)) Let $Q_0(\hat{\xi})$ be a prior distribution over bias corrections and $Q(\hat{\xi}|z)$ be a data-dependent posterior. With the probability $1 - \delta$ over the set $\mathcal{D}^N$, we have*

$$R_{\text{prob}} \leq \frac{1}{N} \sum_{i=1}^{N} \mathbb{E}_{\hat{\xi}\sim Q}\Big[\mathbb{1}\big(h(z_i - \hat{\xi}_i) \neq y_i\big)\Big]$$
$$+ \sqrt{\frac{\text{KL}(Q\,\|\,Q_0) + \ln\left(\frac{2\sqrt{N}}{\delta}\right)}{N}}. \quad (6)$$

The bound highlights that the probabilistic method achieves a balance between fitting the training data (low empirical risk, the first term) and maintaining simplicity (low KL-divergence, the second term). By controlling the trade-off between these two terms, the bound ensures that the model generalizes well to unseen data, even when the output representations are biased.

**Theorem 4.2.** *(Proofs: Appendix A) Based on the theorem above, the probabilistic method achieves a strictly tighter generalization error bound than the deterministic baseline:*

$$R_{\text{prob}} \leq R_{\text{det}} + \mathcal{O}\Big(\sqrt{\frac{\text{KL}(Q\,\|\,Q_0)}{N}}\Big). \quad (7)$$

This theorem demonstrates that the probabilistic method (our ProbSurgery) not only achieves a tighter error bound but also explicitly reduces the generalization error compared to the deterministic baseline (i.e. Surgery).

### 4.3. Extending ProbSurgery to One-to-All Setting

In this section, we move beyond the "one-for-one" constraint in Surgery and propose two strategies to extend a single ProbSurgery module for all merged tasks while preserving competitive performance.

**I: Direct feature alignment (DFA).** Owing to the strong performance of the probabilistic model, we empirically uncover that a single ProbSurgery module alone can handle all biased representation from multiple tasks. Thus, the optimized objective in Eq. (4) is reformulated as

$$\underset{\{\omega\}}{\arg\min}\, \mathcal{L}_{\text{align}} + \lambda \mathcal{L}_{\text{reg}}$$
$$= \underset{\omega}{\arg\min} \sum_{t=1}^{T} \sum_{i=1}^{|D_t^{te}|} \psi(z_{i,t}^{\theta_{\text{unif}}} - \hat{\xi}_i, z_i^{\theta_t}) + \lambda\, L_{\text{KL}}, \quad (8)$$

where the original optimized target $\{\omega_1, ..., \omega_T\}$ is replaced by $\omega$. Experimental results show that existing model merging approaches can achieve better performance improvements even when only a single ProbSurgery module is used, compared to scenarios with multiple Surgery modules.

**II: Proxy distribution alignment (PDA).** To further boost the performance of the probabilistic model in uncertainty prediction, we change the mode of "feature-to-feature" alignment to the "distribution-to-feature" alignment by minimizing *scoring rules* (Gneiting & Raftery, 2007), a function $\mathcal{S}(\cdot)$ that evaluates how well a predicted distribution $P$ over a random variable $\mathcal{X}$ aligns with the actually observed realizations x of $\mathcal{X}$.

To be specific, we change the alignment target from the optimal representation $z^{\theta_t}$ to the bias $z^{\theta_{\text{unif}}} - z^{\theta_t}$. Then, by *scoring rules*, we can calculate the score between the generated bias distribution $Q(\mu, \sigma)$ and the actual observed bias. Eventually, the optimization objective is

$$\underset{\omega}{\arg\min}\, \mathcal{S}(Q_\omega, P^{\text{bias}}) := \underset{\omega}{\arg\min}\, \mathbb{E}_{z^{\text{bias}}\sim P^{\text{bias}}}[Q_\omega, z^{\text{bias}}]. \quad (9)$$

In practical implementation, we have only one realized observation for the bias, i.e., $z^{\text{bias}} = z^{\theta_{\text{unif}}} - z^{\theta_t}$. Then, we approximate $\mathcal{S}(\cdot)$ to two terms and write the objective as

$$\underset{\omega}{\arg\min} \sum_{t=1}^{T} \sum_{i=1}^{|D_t^{te}|} \Big[\frac{\kappa}{r} \sum_{j=1}^{r} \psi(z_{i,t}^{j}, z_{i,t}^{\text{bias}}) + (1-\kappa)\psi(z_{i,t}^{j}, z_{i,t}^{k})\Big], \quad (10)$$

where $\kappa \in [0, 1]$ and is empirically set as 0.5, and $r$ denotes the sampling number that is set as 5. Note that the first term is a function of the sampled bias $z^j$ and the observation bias $z^{\text{bias}}$. The second term is a function of two embeddings, $z^j$ and $z^k$, drawn from the distribution $Q_\omega$. More analyses for the sampling number $r$ can be found in Appendix C.

A detailed comparison of varying methods is shown in Table 1. Experimental results showcase that the performance of

*Table 2.* Multi-task performance when merging ViT-B/32 models on eight vision tasks. The green value denotes the improvement of ProbSurgery compared with the counterpart method, Surgery. Results with more backbones are shown in Appendix C.5.

| Method | SUN397 | Cars | RESISC45 | EuroSAT | SVHN | GTSRB | MNIST | DTD | Avg |
|---|---|---|---|---|---|---|---|---|---|
| Pretrained | 62.3 | 59.7 | 60.7 | 45.5 | 31.4 | 32.6 | 48.5 | 43.8 | 48.0 |
| Individual | 75.3 | 77.7 | 96.1 | 99.7 | 97.5 | 98.7 | 99.7 | 79.4 | 90.5 |
| Traditional MTL | 73.9 | 74.4 | 93.9 | 98.2 | 95.8 | 98.9 | 99.5 | 77.9 | 88.9 |
| Fisher Merging (Matena & Raffel, 2022) | 68.6 | 69.2 | 70.7 | 66.4 | 72.9 | 51.1 | 87.9 | 59.9 | 68.3 |
| RegMean (Jin et al., 2023) | 65.3 | 63.5 | 75.6 | 78.6 | 78.1 | 67.4 | 93.7 | 52.0 | 71.8 |
| Concrete TA (Tang et al., 2023) | 62.5 | 61.1 | 76.0 | 95.7 | 91.0 | 81.9 | 98.5 | 51.9 | 77.3 |
| Concrete AM (Tang et al., 2023) | 67.8 | 70.0 | 87.5 | 96.0 | 91.6 | 96.7 | 98.7 | 63.8 | 84.0 |
| TW AdaMerging (Yang et al., 2024c) | 58.0 | 53.2 | 68.8 | 85.7 | 81.1 | 84.4 | 92.4 | 44.8 | 71.1 |
| Weight Averaging | 65.3 | 63.4 | 71.4 | 71.7 | 64.2 | 52.8 | 87.5 | 50.1 | 65.8 |
|     w/ Surgery (Yang et al., 2024a) | 67.6 | 64.6 | 85.8 | 96.8 | 76.9 | 82.9 | 97.8 | 67.3 | 80.0 |
|     w/ ProbSurgery (Ours) | 70.7 | 70.1 | 94.0 | 99.6 | 83.4 | 98.7 | 99.3 | 78.1 | 86.7 (6.7) |
| Task Arithmetic (Ilharco et al., 2023) | 55.2 | 54.9 | 66.7 | 78.9 | 80.2 | 69.7 | 97.3 | 50.4 | 69.1 |
|     w/ Surgery (Yang et al., 2024a) | 63.8 | 59.9 | 83.3 | 97.9 | 87.0 | 87.0 | 98.6 | 69.4 | 80.9 |
|     w/ ProbSurgery (Ours) | 67.0 | 67.0 | 94.1 | 99.8 | 91.2 | 98.8 | 99.4 | 79.0 | 87.0 (6.1) |
| Ties-Merging (Yadav et al., 2023) | 65.0 | 64.4 | 74.8 | 77.4 | 81.2 | 69.3 | 96.5 | 54.5 | 72.9 |
|     w/ Surgery (Yang et al., 2024a) | 69.8 | 66.1 | 87.3 | 97.5 | 86.7 | 87.6 | 98.5 | 71.6 | 83.1 |
|     w/ ProbSurgery (Ours) | 71.5 | 70.6 | 94.4 | 99.7 | 90.6 | 98.9 | 99.4 | 78.9 | 88.0 (4.9) |
| LW AdaMerging (Yang et al., 2024c) | 64.5 | 68.1 | 79.2 | 93.8 | 87.0 | 91.9 | 97.5 | 59.1 | 80.1 |
|     w/ Surgery (Yang et al., 2024a) | 71.2 | 72.0 | 92.3 | 99.0 | 92.2 | 97.9 | 99.0 | 76.1 | 87.5 |
|     w/ ProbSurgery (Ours) | 71.7 | 73.1 | 94.8 | 99.7 | 93.6 | 98.8 | 99.5 | 79.3 | 88.8 (1.3) |
| **SOTA**: EMR-MERGING (Huang et al., 2024) | 75.2 | 72.8 | 93.5 | 99.5 | 96.9 | 98.1 | 99.6 | 74.4 | 88.7 |

*Table 3.* Multi-task performance when merging BERT models on five NLP tasks. The green value denotes the improvement of ProbSurgery compared with the counterpart method Surgery (Yang et al., 2024a).

| Method | AG News | Yelp Sentiment | Amazon Sentiment | Yahoo Q&A | DBPedia Wikipedia | Avg. |
|---|---|---|---|---|---|---|
| Traditional MTL | 90.6 | 59.1 | 55.6 | 71.3 | 98.5 | 75.0 |
| KD4MTL (Li & Bilen, 2020) | 91.6 | 59.2 | 57.0 | 71.2 | 98.7 | 75.5 |
| Weight Averaging | 79.2 | 49.8 | 45.0 | 50.3 | 55.1 | 55.8 |
|     w/ Surgery | 90.3 | 58.0 | 54.2 | 70.8 | 98.4 | 74.3 |
|     w/ ProbSurgery (Ours) | 91.4 | 62.1 | 55.8 | 72.4 | 98.1 | 76.0 (1.7) |
| Task Arithmetic (Ilharco et al., 2023) | 82.9 | 55.8 | 48.4 | 53.1 | 81.5 | 64.3 |
|     w/ Surgery | 89.8 | 58.4 | 55.4 | 70.3 | 98.0 | 74.4 |
|     w/ ProbSurgery (Ours) | 90.7 | 60.4 | 56.4 | 72.3 | 98.7 | 75.7 (1.3) |

only one single ProbSurgery module with DFA (or PDA) can remarkably outperform that of multiple Surgery modules.

## 5. Experiments

### 5.1. Experimental Setup

All experiments are implemented by `Pytorch` library and conducted on a single NVIDIA RTX A6000. We tested three times with different random seeds and reported the average performance.

We categorize the compared methods into two parts: 1) *basic (non-merged) methods*: Pretrained, Individual, and Traditional MTL, 2) *model merging methods*: Weight Averaging, Fisher Merging (Matena & Raffel, 2022), RegMean (Jin et al., 2023), Task Arithmetic (Ilharco et al., 2023), Ties-Merging (Yadav et al., 2023), Concrete TA (Tang et al.,

2023), Concrete AM (Tang et al., 2023), AdaMerging (Yang et al., 2024c), Surgery (Yang et al., 2024a), and EMR-MERGING (Huang et al., 2024).

**Architecture.** (1) Following Surgery (Yang et al., 2024a), we adopt CLIP (Radford et al., 2021) as the backbone model for vision tasks, where various ViT family architectures are utilized for the visual encoder. For NLP tasks, we assign Bert as the backbone. (2) The structure of ProbSurgery is a lightweight, three-layer connected MLP network, formally denoted by $\{h_1, h_2, h_1\}$, where $h_1$ represents the embedding size output by the encoder in CLIP or NLP models and $h_2$ is the width of the hidden layer, a hyperparameter (set as 128 for all settings) in this paper.

Other details are divided into three parts, including 1) **Datasets.** See Appendix B.1, 2) **Baseline.** See Appendix B.2, and 3) **Training implementation.** See Appendix B.3.

*Table 4.* Comparison results of ProbSurgery and Surgery. ✗ denotes "learning eight modules for eight tasks independently" and ✓ denotes "learning one module for all merged tasks".

| Methods | One Module | SUN397 | Cars | RESISC45 | EuroSAT | SVHN | GTSRB | MNIST | DTD | Avg. |
|---|---|---|---|---|---|---|---|---|---|---|
| Task Arithmetic | | 55.2 | 54.9 | 66.7 | 78.9 | 80.2 | 69.7 | 97.3 | 50.4 | 69.1 |
| w/ Surgery | ✗ | 63.8 | 59.9 | 83.3 | 97.9 | 87.0 | 87.0 | 98.6 | 69.4 | 80.9 |
| **w/ ProbSurgery** | ✗ | 67.0 | 67.0 | 94.1 | 99.8 | 91.2 | 98.8 | 99.4 | 79.0 | 87.0 |
| w/ Surgery | ✓ | 60.3 | 56.0 | 70.1 | 92.1 | 82.7 | 76.8 | 97.7 | 55.0 | 73.8 |
| **w/ ProbSurgery** | ✓, **DFA** | 62.2 | 58.4 | 84.8 | 98.9 | 88.7 | 94.2 | 98.8 | 68.4 | 81.8 |
| **w/ ProbSurgery** | ✓, **PDA** | 63.1 | 58.7 | 85.7 | 99.5 | 89.0 | 95.0 | 99.2 | 68.9 | 82.4 |

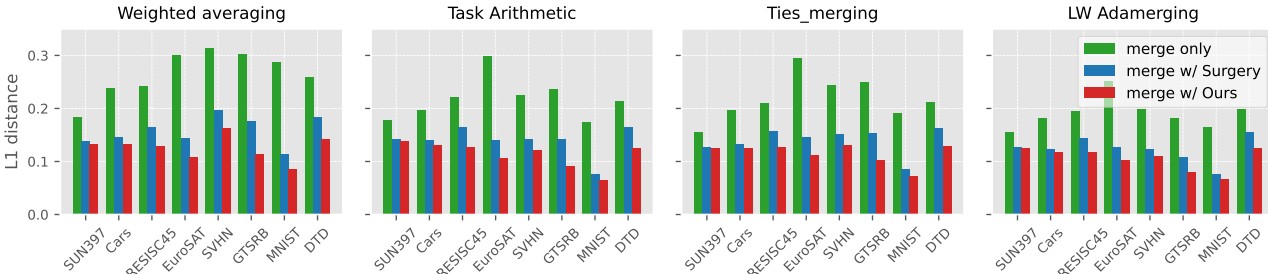

*Figure 4.* Performance comparisons of our ProbSurgery with Surgery (Yang et al., 2024a) regarding calibrating the representation bias.

## 5.2. Main Results

**Vision tasks.** In Table 2, we report the comparison results of ProbSurgery with previous state-of-the-art methods on eight vision tasks where the backbone is ViT-B/32. We can observe that 1) current model merging methods are significantly worse than traditional multi-task learning methods, even for the best one, LW adamerging. It is because the issue of parameter interference largely degrades the task knowledge provided by different models. 2) ProbSurgery demonstrates consistent improvements across various model merging strategies, including all four base methods, achieving notably higher average accuracy on a diverse set of vision tasks. Compared to standard Surgery, ProbSurgery delivers stronger and more stable gains without sacrificing performance on individual tasks. For example, applying ProbSurgery to Weight Averaging boosts the average accuracy from 80.0% to 86.7%, an increase of 6.7%. These consistent gains across different model merging strategies highlight the robust performance benefits that ProbSurgery brings to multi-task learning.

**NLP tasks.** Following (Yang et al., 2024a), we adopt Bert as the backbone for five NLP tasks and then conduct model merging. We report the results in Table 3. From this table, our post-calibration method (ProbSurgery) consistently outperforms Surgery on all five NLP tasks, yielding up to a 1.7% improvement (from 74.3% to 76.0%) under Weight Averaging and a 1.3% improvement (from 74.4% to 75.7%) under Task Arithmetic. Notably, this is the first time a post-hoc calibration approach has surpassed a dedicated multi-task learning algorithm (KD4MTL (Li & Bilen, 2020)), as

ProbSurgery achieves an average accuracy of 76.0% versus KD4MTL's 75.5%. These results underscore the effectiveness and broad applicability of our method in delivering state-of-the-art performance across diverse NLP tasks.

## 5.3. Performance in *One-to-All* Setting

Ideally, we hope to resort to one module to rectify representations from all merged tasks, which greatly reduces the training cost and improves the practicality. In Table 4, the results demonstrate the superiority of our proposed ProbSurgery approach. Notably, using a single ProbSurgery module significantly outperforms the performance achieved by using eight independent Surgery modules (81.8% or 82.4% *vs.* 80.9%), as reflected in the average accuracy scores. Additionally, both strategies, DFA and PDA, consistently enhance performance across multiple tasks, further validating the effectiveness of our probabilistic framework. This highlights the efficiency and robustness of our method in handling diverse tasks with a unified module, largely promoting the real-world application of ProbSurgery.

## 5.4. More Analysis

**Performance on modeling the representation bias.** In this work, we aim to minimize the discrepancy in the representation distribution between the merged and individual models. To demonstrate this, we plot L1 distance after post-calibrating four methods with our ProbSurgery.

The results are presented in Figure 4. Overall, our method (represented by the red bars) consistently achieves lower L1 distance across various tasks and base methods, indi-

*Table 5.* Comparison results of ProbSurgery and Surgery in the setting of *domain shift* and *out-of-distribution*.

| Settings | Domain shift | | | | Out-of-distribution | | | |
|---|---|---|---|---|---|---|---|---|
| Methods | Weighted Averaging | | Task Arithmetic | | Weighted Averaging | | Task Arithmetic | |
| | In-domain | Other-domain | In-domain | Other-domain | ACC (↑) | AUROC (↑) | ACC (↑) | AUROC (↑) |
| Only merge | 84.6 | 71.4 | 86.2 | 73.1 | 82.7 | 58.5 | 84.2 | 62.1 |
| **w/ Surgery** | 89.1 (4.5) | 72.1 (0.7) | 90.2 (4.0) | 74.6 (1.5) | 90.6 (7.9) | 69.5 (11.0) | 92.7 (8.5) | 72.6 (10.5) |
| **w/ ProbSurgery** | 91.4 (6.8) | 76.8 (5.4) | 93.1 (6.9) | 77.2 (4.1) | 93.7 (11.0) | 76.1 (17.6) | 95.5 (11.3) | 80.2 (18.1) |

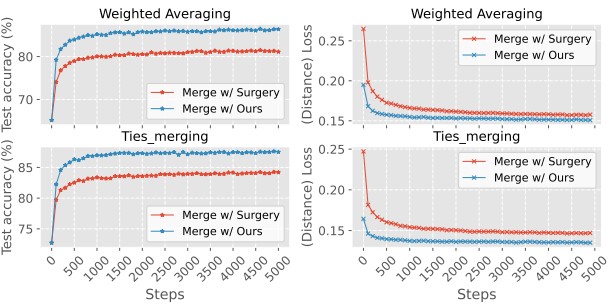

*Figure 5.* Better efficiency is achieved by our proposed Prob-Surgery compared to Surgery (Yang et al., 2024a).

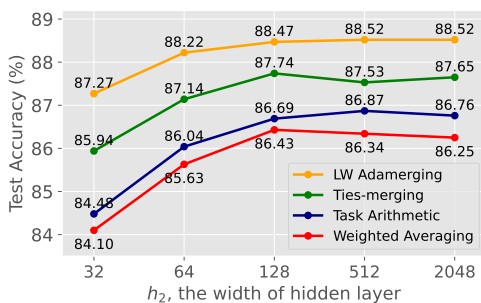

*Figure 6.* Hyper-parameter selection for $h_2$, which denotes the width of the hidden layer in ProbSurgery module.

cating superior performance in mitigating representation bias. Compared with "merge only" (green) and "merge w/ surgery" (blue), our approach not only reduces L1 distance more significantly in most tasks but also demonstrates consistent effectiveness regardless of different merging strategies. This highlights the stability and general applicability of our method, which can better align the merged representations with their expected distributions, ultimately offering improved generalization in multi-task settings. More results on various distance metrics are shown in Appendix D.

**Generalization.** To evaluate the generalization of Prob-Surgery, we conduct experiments on two settings, including:

- **Domain shift.** We test the performance in a domain-shifted setting, which would reflect the generalization of generated representation bias. Specifically, we merge four individual models, independently trained on *SUN397, Cars, HomeOffice (Real-World), and DTD* with traditional model merging methods. After post-calibrating, we compute the average test accuracy on the other three domain datasets - *HomeOffice (Art, Clipart, and Product).* The results are shown in Table 5 *left*, reflecting that the improvement on the in-domain dataset brought by Surgery can hardly transfer to other-domain datasets. In detail, we can see that with weighted averaging, the improvement of test accuracy in *Real-world* domain achieves 4.5% after training with Surgery, while only an average 0.7% improvement is obtained in the other three domains. In contrast, we have a performance increase of 5.4%.

- **Out-of-distribution.** We compare ProbSurgery with Surgery on the OOD Detection task, where the test phase will occur with samples from other datasets/distributions.

We merged four models trained on *CIFAR-10, EuroSAT, SVHN, and GTSRB*. Then, we introduce 500 samples from SUN397 in the test phase as the OOD data. More experimental details can be found in Appendix B.5. In Table 5 *right*, ProbSurgery outperforms Surgery on OOD tasks by achieving higher ACC and AUROC in both methods. For instance, it boosts ACC from 90.6% to 93.7% and AUROC from 69.5% to 76.1% under Weighted Averaging.

Overall, we can consider that the variance in our probabilistic manner introduces extra cross-model information, contributing to boosting the generalization performance.

**Efficiency.** Probabilistic models often achieve faster convergence than deterministic models because they effectively model and capture uncertainty, which reduces the risk of overfitting and leads to smoother optimization paths.

In Figure 5, we plot the test accuracy of four methods integrated with Surgery (Yang et al., 2024a) and ours. From these curves, it is evident that our method ProbSurgery (blue lines) converges much faster than Surgery (red lines) on all merging methods. With only 300 steps, ProbSurgery has already surpassed Surgery's best performance across all four base methods. This improvement underscores the efficiency of our technique, demonstrating that even with fewer training steps, it can achieve superior test accuracy.

### 5.5. Ablation Study

**Hyper-parameters.** In ProbSurgery, there exists one essential hyper-parameter, i.e., $h_2$, the hidden layer's width. Results shown in Figure 6 demonstrate that the choice of $h_2$ would impact test accuracy. The best performance among

all base methods is observed when the value of $h_2$ is between 128 and 512. At $h_2 = 128$, the hidden layer is wide and the complexity of learnable parameters is enough to capture complex patterns, leading to a significant accuracy boost. Increasing $h_2$ to 512 further stabilizes performance, but beyond this point (e.g., $h_2 = 2048$), accuracy slightly decreases due to overfitting caused by excessive capacity. Thus, selecting $h_2$ within the interval of $[128, 512]$ contributes to both model generalization and task-specific learning.

## 6. Conclusion

In this paper, we follow one research - Surgery (Yang et al., 2024a) and study how to mitigate the representation bias inherent in the final merged model, which aims to fill the performance gap between the merged model and multiple individual models. To this end, we propose to model the bias in a probabilistic manner and design a module named Prob-Surgery. This module outputs more robust representation bias via sampling from the generated distribution that well models the uncertainty within model merging. Meanwhile, we propose two strategies that can extend ProbSurgery to the one-to-all post-calibration setting, which brings more practical values. Extensive experiments demonstrate the effectiveness of ProbSurgery.

## Acknowledgment

This research is supported by the Ministry of Education, Singapore, under its MOE AcRF Tier 3 Award MOE-MOET32022-0006 and MOE AcRF Tier 2 Award T2EP20223-0022. Any opinions, findings and conclusions or recommendations expressed in this material are those of the author(s) and do not reflect the views of the Ministry of Education, Singapore.

## Impact Statement

This paper aims to correct biased representations resulting from model merging, a promising approach to multi-task learning. There are many potential societal consequences of our work, none of which we feel must be specifically highlighted here.

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

## Organization of the Supplemental Material

. The main contents of this appendix are as follows:

- Appendix A: We provide the theoretical proofs for the given therome in the paper.
- Appendix B: We describe the datasets, training procedures, and architecture of PorbSurgery in detail.
- Appendix C: We show some experimental results and analyses that are deleted due to the page limit of the main text.
- Appendix D: We visualize the representation generated by Surgery and ProbSurgery, respectively, via t-SNE technique (Van der Maaten & Hinton, 2008).

## A. Theoretical Proofs

In this section, we first provide an analysis for the limitation of the deterministic method and then give detailed proof steps for Theorem 4.2.

### A.1. Analysis for the Deterministic Method

The deterministic method as a post-representation calibration manner, like surgery, is equivalent to a Dirac delta posterior:

$$Q_{\text{det}}(\xi|\boldsymbol{z}) = \delta(\xi - \xi_{\text{det}}), \quad \text{where } \delta(\xi - \xi_{\text{det}}) = \begin{cases} +\infty, & \xi = \xi_{\text{det}} \\ 0, & \xi \neq \xi_{\text{det}} \end{cases} \tag{11}$$

Substituting into the PAC-Bayes bound (McAllester, 2003), the KL-divergence term $\text{KL}(Q_{\text{det}} \| Q_0)$ diverges unless $Q_0$ is also a delta distribution, which violates the PAC-Bayes assumptions. Thus, deterministic representation correction cannot leverage PAC-Bayes bounds and instead relies on classical VC-dimension bounds:

$$R_{\text{det}} \leq \hat{R}_{\text{det}} + \mathcal{O}(\sqrt{\frac{d}{N}}), \tag{12}$$

where $d$ denotes the complexity of the three-layer MLP in Surgery, and $R_{\text{det}}, \hat{R}_{\text{det}}$ denote the expected and empirical risk, respectively.

### A.2. Proofs for Theorem 4.2

From Theorem 4.1, for any prior $Q_0$ and posterior $Q$, with the probability $1 - \delta$ over the training set $\mathcal{D}^N$, we have

$$R_{\text{prob}} \leq \underbrace{\frac{1}{N} \sum_{i=1}^{N} \mathbb{E}_{\xi \sim Q} \left[ \mathbf{1}(h(z_i - \xi) \neq y_i) \right]}_{\text{Empirical Risk } \hat{R}_{\text{prob}}} + \underbrace{\sqrt{\frac{\text{KL}(Q \| Q_0) + \ln \frac{2\sqrt{N}}{\delta}}{N}}}_{\text{Complexity Term}} . \tag{13}$$

For the first term, i.e., the empirical risk term for ProbSurgery, it satisfies:

$$\frac{1}{N} \sum_{i=1}^{N} \mathbb{E}_{\xi \sim Q} \left[ \mathbb{1}(h(z_i - \xi) \neq y_i) \right] \leq \frac{1}{N} \sum_{i=1}^{N} \mathbb{1}(h(z_i - \xi_{\text{det}}) \neq y_i), \tag{14}$$

where $\xi_{\text{det}} = \mathbb{E}_{\xi \sim Q}[\xi]$ is the deterministic correction (equivalent to Surgery). This follows from Jensen's inequality applied to the convex 0-1 loss. Thus, we can deduce that $\hat{R}_{\text{prob}} \leq \hat{R}_{\text{det}}$.

For the deterministic method (Surgery), classical VC-dimension bounds give:

$$R_{\text{det}} \leq \hat{R}_{\text{det}} + \mathcal{O}\left( \sqrt{\frac{d}{N}} \right), \tag{15}$$

where $d$ is the VC-dimension (or parameter count) of the deterministic network.

Eventually, by substituting the empirical risk bound in Eq. (14) into the PAC-Bayes bound in Eq. (13) and then combining with Eq. (15), we have

$$R_{\text{prob}} \leq R_{\text{det}} + \mathcal{O}\Big(\sqrt{\frac{\text{KL}(Q \,\|\, Q_0)}{N}}\Big). \tag{16}$$

Here, we omit the item of $\ln \frac{2\sqrt{N}}{\delta}$ since its value is very small given a large scale of the training set (i.e., $N$ is large).

## B. Experimental Setup

### B.1. Datasets

Following prior studies on model merging, such as Task Arithmetic (Ilharco et al., 2023), Ties-Merging (Yadav et al., 2023), AdaMerging (Yang et al., 2024c), and Surgery (Yang et al., 2024a), we merge models trained on the following eight vision datasets and five NLP datasets:

- **SUN397** (Xiao et al., 2016): A benchmark for Scene Understanding (SUN) featuring 108,753 images across 397 classes. Each class contains at least 100 images.

- **Cars** (Krause et al., 2013): The Stanford Cars dataset comprises 16,185 images spanning 196 car classes, approximately split 1:1 into training and test sets.

- **RESISC45** (Cheng et al., 2017): A publicly available benchmark for remote sensing image scene classification. It includes 45 scene classes, each with 700 images (256×256 resolution), totaling 31,500 images.

- **EuroSAT** (Helber et al., 2019): A dataset of Sentinel-2-based satellite images focusing on land-use classification. It consists of 27,000 labeled and geo-referenced images divided into 10 classes.

- **SVHN** (Yuval, 2011): A dataset of real-world house number images in 10 classes, visually resembling MNIST (LeCun, 1998) but containing over 600,000 color images.

- **GTSRB** (Stallkamp et al., 2011): The German Traffic Sign Recognition Benchmark, with over 50,000 images in 43 classes. Images vary in lighting conditions and backgrounds.

- **MNIST** (LeCun, 1998): One of the most renowned datasets in machine learning: 70,000 (60k training + 10k testing) grayscale images of handwritten digits in 10 classes, each sized 28×28.

- **DTD** (Cimpoi et al., 2014): The Describable Textures Dataset (DTD) comprises 5,640 labeled texture images in 47 classes, with each class containing around 120 images (ranging from 300×300 to 640×640 in resolution).

- **AGNews** (Del Corso et al., 2005): A news classification dataset with roughly 120k training samples and 7.6k test samples, covering 4 classes (World, Sports, Business, Science/Technology).

- **Yelp** (Zhang et al., 2015): A sentiment analysis dataset of user-submitted reviews (positive vs. negative), containing around 560k training samples and 38k test samples.

- **Amazon** (Zhang et al., 2015): Another sentiment analysis dataset (positive vs. negative) from Amazon product reviews, including about 3.6M training samples and 400k test samples.

- **DBPedia** (Zhang et al., 2015): Focused on categorizing Wikipedia article excerpts into 14 classes (e.g., Company, Artist, EducationalInstitution), with approximately 560k training samples and 70k test samples.

- **Yahoo** (Auer et al., 2007): A question-and-answer classification dataset of around 1.4 million training samples and 60k test samples, covering 10 classes (e.g., Society & Culture, Science & Mathematics, Health).

### B.2. Baselines

We categorize the compared methods into two parts: 1) *basic (non-merged) methods*: Pretrained, Individual, and Traditional MTL, 2) *model merging methods*: includes as follows

- **Weight Averaging:** Simply averages the corresponding weights from multiple trained models to a merged model.

- **Fisher Merging** (Matena & Raffel, 2022): Merges models by weighting parameters according to Fisher information, prioritizing directions crucial for each task.

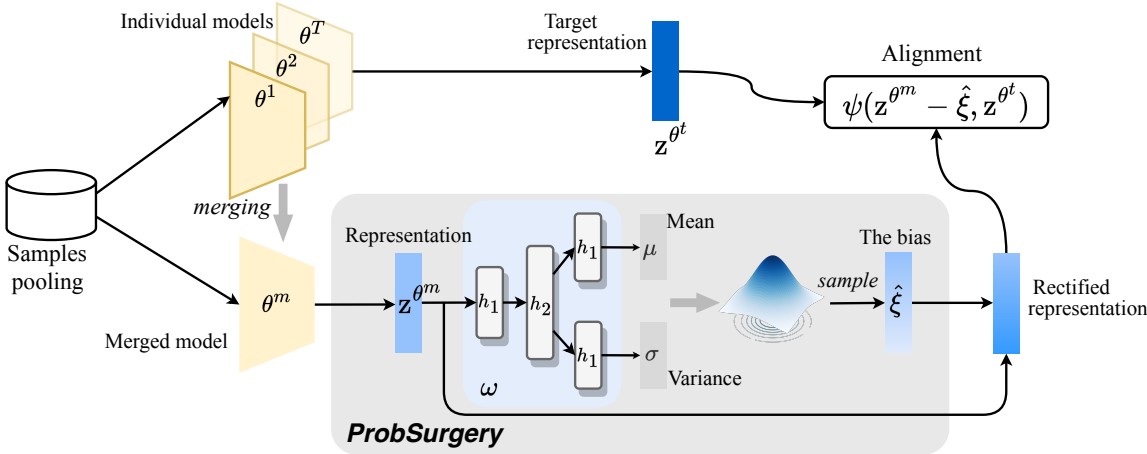

*Figure 7.* Flowchat of the post-representation calibration processes with our proposal, ProbSurgery.

*Table 6.* Comparison of **the number of learnable parameters** within Surgery and ProbSurgery. Our proposal ProbSurgery can be applied to *one-to-all* settings, i.e., learning one ProbSurgery module for all merged tasks. Thus, the number of extra parameters will not increase with the number of merged tasks increase.

| Networks | ViT-B/16 | ViT-B/32 | ViT-L/14 |
|---|---|---|---|
| Number of the backbone's parameters | $907,589,640$ | $894,337,032$ | $2,740,496,392$ |
| Number of extra learnable parameters Surgery Module w/ merging $N$ tasks | | $131,072$ $N \times 131,072$ | $196,608$ $N \times 196,608$ |
| ProbSurgery Module w/ merging $N$ tasks | | $131,712$ $131,712$ | $197,504$ $197,504$ |

- **RegMean** (Jin et al., 2023): Adds a regularization term when averaging models, penalizing large deviations from each individual task model.

- **Task Arithmetic** (Ilharco et al., 2023): Views models as points in parameter space and combines them via vector arithmetic operations (e.g., addition, subtraction).

- **Ties-Merging** (Yadav et al., 2023): Selectively ties and reuses shared parameters across models to preserve beneficial components during merging.

- **Concrete TA & AM** (Tang et al., 2023): Provides a continuous relaxation of task arithmetic, learning an optimal linear mixture of task-specific parameters.

- **AdaMerging** (Yang et al., 2024c): Adaptively reweights each model's parameters based on their contribution to the merged solution.

### B.3. Implementation Details

To (post)train and learn the parameters $\omega$ of our proposal, ProbSurergy, we follow the convention in Surgery (Yang et al., 2024a). To be specific, we do not require any information from the test labels rather than adopting a self-training manner. For training, we adopt Adam as the optimizer with a learning rate of $1 \times 10^{-3}$ for all training iterations. We totally train the ProbSurgery module for $5,000$ iteration with a batch size of 16. During the training phase, we utilize the stochastic sampling strategy, i.e., the reparameterization trick, to obtain the generated representation bias. For inference and testing, we consider the mean value $\mu$ output by ProbSurgery as the representation bias.

*Table 7.* Multi-task performance on the ViT-B/32 model when **different loss functions** are used in the representation surgery module. The green value denotes the improvement of ProbSurgery compared with the counterpart method Surgery (Yang et al., 2024a).

| Methods | Loss Func. | SUN397 | Cars | RESISC45 | EuroSAT | SVHN | GTSRB | MNIST | DTD | **Avg.** |
|---|---|---|---|---|---|---|---|---|---|---|
| Task Arithmetic (Ilharco et al., 2023) | | 55.2 | 54.9 | 66.7 | 78.9 | 80.2 | 69.7 | 97.3 | 50.4 | 69.1 |
| **w/ Surgery** | (L1 Loss) | 63.8 | 59.9 | 83.3 | 97.9 | 87.0 | 87.0 | 98.6 | 69.4 | 80.9 |
| **w/ ProbSurgery** | (L1 Loss) | 66.3 | 65.8 | 94.3 | 99.7 | 90.5 | 98.6 | 99.4 | 78.0 | 86.6 (5.7) |
| **w/ Surgery** | (MSELoss) | 64.3 | 59.8 | 84.0 | 97.8 | 87.6 | 88.7 | 98.8 | 69.8 | 81.4 |
| **w/ ProbSurgery** | (MSELoss) | 66.5 | 65.9 | 94.1 | 99.7 | 91.1 | 98.9 | 99.5 | 78.5 | 86.7 (5.3) |
| **w/ Surgery** | (SmoothL1Loss) | 64.1 | 59.7 | 84.1 | 97.9 | 88.1 | 89.7 | 98.8 | 70.6 | 81.6 |
| **w/ ProbSurgery** | (SmoothL1Loss) | 66.0 | 63.4 | 93.9 | 99.6 | 90.8 | 98.5 | 99.4 | 78.2 | 86.2 (4.6) |

## B.4. Architecture of ProbSurgery

Our proposal module, ProbSurgery, belongs to the lightweight network, which is parameterized by a three-layer fully connected MLP and exhibited in Figure 7. We can observe that this module is simple and contains fewer learnable parameters, whose architecture is represented by $\{h_1, h_2, h_1\}$. Note that $h_1$ denotes the width of feature embedding output by the encoder in CLIP or NLP models. For example, $h_1$ is set as 512 for ViT-B/16, 32 and 768 for ViT-L/14. Furthermore, $h_2$ is the width of the hidden layer, a hyperparameter in this paper. We set $h_2 = 128$ for all experiments.

We report the parameter comparison between Surgery and our ProbSurgery in Table 6. Firstly, even though our method belongs to a probabilistic manner, it does not significantly introduce more parameters compared with Surgery. Then, when the number of merged tasks increases, the advantage of the learnable parameter would be more obvious since ours can be directly extended into *one-to-all* settings.

## B.5. Implementation of OOD Experiments

In the OOD experiment, we introduce two metrics, i.e., test accuracy and AUROC, to evaluate the performance of Surgery and ProbSurgery. The computation details for these two metrics are as follows. When we finish the model merging and the post-calibration process with (Prob)Surgery, we test the performance of different model merging methods on OOD detection. We use the test sets of four tasks and a small OOD set of 500 samples randomly sampled from the test set of SUN397, denoted by $D_0, D_1, D_2, D_3$, and $D_o$.

To measure **test accuracy**, we follow the standard model-merging approach and maintain separate task-specific classifiers for the merged tasks. Formally, the test accuracy on the test set of the $t$-th task $D_t$ can be expressed as $ACC_t = \frac{\sum_{(x,y)\in D_t+D_o} \mathbb{1}(\hat{y}=y)}{\|D_t\|+\|D_o\|}$, where $\hat{y}$ denotes the predicted label.

To report **AUROC**, we use a commonly adopted unsupervised technique that does not rely on the classifier. Specifically, for each task, we store its class prototypes during training (when no OOD samples are present). In the test phase, for task $t$, we compute the representation of every sample in the test set $D_t + D_o$ and measure its distance to the stored class prototypes. The greater the distance, the more likely the sample is to be an outlier. Eventually, the AUROC value can be calculated on the distance and the 0-1 label (where 0 and 1 denote the In-distribution and Out-of-distribution samples, respectively).

# C. More Experimental Analyses

## C.1. Impacts of Loss Functions

In this paper, to measure the discrepancy in the representation distribution be- tween the merged and individual models, we utilize *L1 distance* (`torch.nn.L1Loss()`) as the optimization loss function, which keeps the same as Surgery (Yang et al., 2024a). To verify that this discrepancy is agnostic with different distance metrics, we test totally four distance metrics and set them as the loss function for comparison, including `L1 loss`, `MSE loss` and `Smooth L1 loss`.

The experimental results in Table 7 demonstrate the superior performance of our proposed method, ProbSurgery, compared to the baseline Surgery across various loss functions. ProbSurgery achieves consistent improvements in average performance, with gains ranging from +4.6 to +5.7, highlighting its adaptability to different loss functions. Notably, ProbSurgery excels in datasets like RESISC45 and DTD, where it significantly reduces task interference and enhances representation alignment,

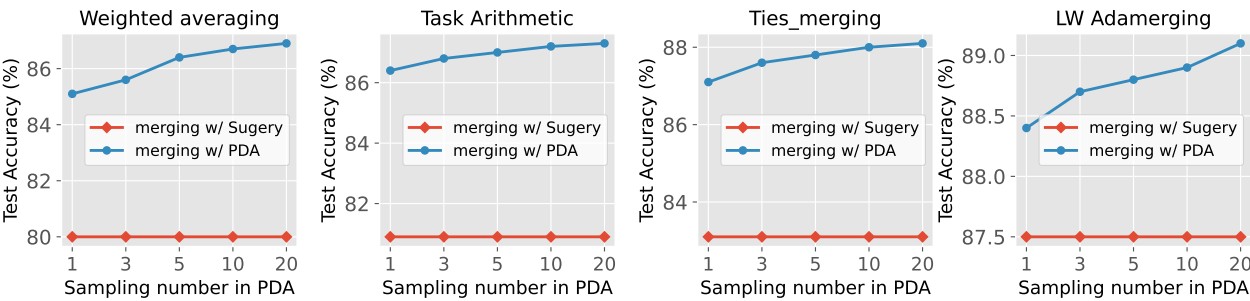

*Figure 8.* Performance impact of the sampling number regarding computing *scoring values* in our Proxy Distribution Alignment (PDA).

achieving gains as high as +11.0 and +8.6, respectively. These improvements demonstrate its robustness in handling diverse tasks, from large-scale datasets like SUN397 to texture-based datasets like DTD. Moreover, its compatibility with different loss functions and ability to generalize across tasks establish ProbSurgery as an effective approach for multi-task learning.

### C.2. Impact of Sampling Number in Proxy Distribution Alignment (PDA)

In real implementation, to better estimate the discrepancy between the predicted distribution $Q_\omega$ with our practical representation bias observation $z^{bias}$, the scoring rules should leverage the sampling strategy (i.e., the reparameterization trick) to compute the distance between two vectors. Therefore, a larger number for sampling can better approximate the true distribution and reduce estimation uncertainty. It is worthy-noted that increasing the sampling number would not raise the learnable parameters and training costs since it does not involve extra gradient computation.

In Figure 8, we report the influence of sampling number regarding computing *scoring values* in our proposed Proxy Distribution Alignment (PDA). The experimental results demonstrate that increasing the sampling number in PDA consistently improves test accuracy across different merging strategies, with the performance gains becoming stable as the sampling number reaches 10 or more. This highlights the effectiveness of leveraging larger sampling sizes to better approximate the true distribution and reduce estimation uncertainty. Notably, the use of PDA consistently outperforms the baseline (merging without PDA) in all methods, with LW Adamerging achieving the highest overall accuracy. Furthermore, the approach introduces minimal computational overhead, as no additional learnable parameters or gradient computations are required, making PDA a practical and efficient method for improving model performance.

### C.3. Impact of the Size of the Validation (Test) Set

In Surgery and ProbSurgery, we rely on unlabeled test/validation data to build self-supervision signals and update the parameters of the (Prob)Surgery module. Here, we made an experiment to verify how the size of the validation set impacts ProbSurgery's performance. In the following table, we can see that using a larger proportion of the validation set helps the ProbSurgery module better capture and mitigate inherent biases, leading to higher overall accuracy. Notably, even when only 10% of the unlabeled validation data is used, our method still exceeds Surgery's performance of 80.9%, demonstrating its robustness under limited data conditions.

*Table 8.* Comparison of performance across datasets under different validation set ratios. The backbone is ViT-B/32.

| Method | Ratio | SUN397 | Cars | RESISC45 | EuroSAT | SVHN | GTSRB | MNIST | DTD | Avg |
|---|---|---|---|---|---|---|---|---|---|---|
| Task Arithmetic | - | 55.2 | 54.9 | 66.7 | 78.9 | 80.2 | 69.7 | 97.3 | 50.4 | 69.1 |
| w/ Ours | 10% | 63.5 | 64.5 | 89.6 | 94.9 | 88.4 | 89.4 | 98.4 | 74.5 | 82.9 |
| w/ Ours | 50% | 65.9 | 66.5 | 92.3 | 97.8 | 90.5 | 95.0 | 98.8 | 77.2 | 85.5 |
| w/ Ours | 100% | 67.0 | 67.0 | 94.1 | 99.8 | 91.2 | 98.8 | 99.4 | 79.0 | 87.0 |
| w/ Surgery | 100% | 63.8 | 59.9 | 83.3 | 97.9 | 87.0 | 87.0 | 98.6 | 69.4 | 80.9 |

## C.4. Performance on Merging Tens of Tasks

To verify the performance of our ProbSurgery in more challenging settings, we additionally add three tasks (ImageNet100, CIFAR100, and real-world Homeoffice) to original eight tasks. The results in Table 9 show that ProbSurgery consistently outperforms both the Task Arithmetic and Surgery baselines across all datasets. Notably, the performance gains are especially pronounced on the newly introduced, more complex benchmarks: ProbSurgery achieves 74.8% on ImageNet100 (vs. 11.6% for Task Arithmetic and 56.8% for Surgery). These results indicate even when merging a large number of models, our method can effectively correct the representation bias and achieve superior performance.

| Method | SUN397 | Cars | RESISC45 | EuroSAT | SVHN | GTSRB | MNIST | DTD | ImageNet100 | CIFAR100 | HomeOffice | Avg |
|---|---|---|---|---|---|---|---|---|---|---|---|---|
| Task Arithmetic | 31.8 | 39.3 | 49.6 | 60.3 | 74.4 | 60.4 | 95.8 | 38.4 | 11.6 | 42.6 | 70.3 | 52.3 |
| w/ Surgery | 50.5 | 52.7 | 76.8 | 96.3 | 83.8 | 79.9 | 98.6 | 65.8 | 56.8 | 55.6 | 80.3 | 72.5 |
| w/ ProbSurgery | 53.7 | 55.7 | 86.3 | 97.9 | 85.5 | 96.0 | 98.8 | 72.9 | 74.8 | 63.4 | 86.2 | **79.2** |

*Table 9.* Performance comparison of Surgery and ProbSurgery on merging 11 tasks.

## C.5. Performance on More Backbones

For vision tasks, we further conduct comparison experiments on a different vision backbone, i.e., ViT-L-14, to verify the effectiveness of our proposal, ProbSurgery.

- **ViT-L/14.** Table 10 demonstrates the performance of various methods for merging ViT-L/14 models across eight tasks. Our proposed method, ProbSurgery, consistently outperforms the baseline Surgery and other existing methods in terms of average performance (Avg.). Notably, ProbSurgery achieves significant improvements, with performance gains ranging from +1.5% to +5.1% over Surgery. This highlights its superior ability to balance task-specific features and enhance overall stability. Moreover, ProbSurgery adapts seamlessly to different merging strategies, such as Weight Averaging, Task Arithmetic, and LW AdaMerging, consistently yielding better results. These advantages make ProbSurgery a highly effective and generalizable solution for multi-task learning and large model merging tasks.

- **ViT-B/16.** Table 11 highlights the superior performance of our proposed method, ProbSurgery, in merging ViT-B/16 models across eight tasks. Compared to the baseline Surgery, ProbSurgery consistently achieves higher average scores, with improvements ranging from +2.4% to +6.9%, demonstrating its ability to balance task-specific features effectively. It excels across diverse datasets, including challenging tasks like DTD and SVHN, and outperforms other methods, such as Weight Averaging, Task Arithmetic, and LW AdaMerging, across all strategies. ProbSurgery's robust and generalizable approach ensures better task balance, enhanced generalization, and improved stability, making it a highly effective solution for multi-task learning.

*Table 10.* Multi-task performance when merging ViT-L/14 models on eight tasks. The green value denotes the improvement of ProbSurgery compared with the counterpart method Surgery (Yang et al., 2024a).

| Method | SUN397 | Cars | RESISC45 | EuroSAT | SVHN | GTSRB | MNIST | DTD | Avg. |
|---|---|---|---|---|---|---|---|---|---|
| Pretrained | 66.8 | 77.7 | 71.0 | 59.9 | 58.4 | 50.5 | 76.3 | 55.3 | 64.5 |
| Individual | 82.3 | 92.4 | 97.4 | 100 | 98.1 | 99.2 | 99.7 | 84.1 | 94.2 |
| Traditional MTL | 80.8 | 90.6 | 96.3 | 96.3 | 97.6 | 99.1 | 99.6 | 84.4 | 93.5 |
| Fisher Merging (Matena & Raffel, 2022) | 69.2 | 88.6 | 87.5 | 93.5 | 80.6 | 74.8 | 93.3 | 70.0 | 82.2 |
| RegMean (Jin et al., 2023) | 73.3 | 81.8 | 86.1 | 97.0 | 88.0 | 84.2 | 98.5 | 60.8 | 83.7 |
| Concrete TA (Tang et al., 2023) | 74.6 | 86.2 | 89.0 | 96.7 | 93.6 | 93.4 | 99.1 | 66.9 | 87.4 |
| Concrete AM (Tang et al., 2023) | 77.8 | 91.2 | 92.1 | 97.0 | 94.4 | 97.9 | 99.0 | 79.5 | 91.1 |
| Weight Averaging | 72.1 | 81.6 | 82.6 | 91.9 | 78.2 | 70.7 | 97.1 | 62.8 | 79.6 |
| w/ **Surgery** (Yang et al., 2024a) | 73.7 | 83.9 | 92.0 | 98.4 | 82.4 | 86.3 | 98.7 | 71.9 | 85.9 |
| w/ **ProbSurgery (Ours)** | 75.7 | 87.1 | 96.5 | 99.6 | 88.9 | 99.0, | 99.5 | 81.6 | 91.0 (5.1) |
| Task Arithmetic (Ilharco et al., 2023) | 73.9 | 82.1 | 86.6 | 94.1 | 87.9 | 86.7 | 98.9 | 65.6 | 84.5 |
| w/ **Surgery** (Yang et al., 2024a) | 75.7 | 84.4 | 93.1 | 98.8 | 91.3 | 93.4 | 99.1 | 76.1 | 89.0 |
| w/ **ProbSurgery (Ours)** | 77.0 | 87.5 | 96.2 | 99.7 | 94.2 | 99.1 | 99.4 | 81.8 | 92.0 (4.0) |
| Ties-Merging (Yadav et al., 2023) | 76.5 | 85.0 | 89.3 | 95.7 | 90.3 | 83.3 | 99.0 | 68.8 | 86.0 |
| w/ **Surgery** (Yang et al., 2024a) | 76.5 | 85.9 | 93.7 | 99.2 | 89.7 | 92.0 | 99.1 | 78.1 | 89.3 |
| w/ **ProbSurgery (Ours)** | 77.7 | 88.4 | 96.7 | 99.7 | 93.2 | 99.0 | 99.5 | 82.4 | 92.6 (3.3) |
| LW AdaMerging (Yang et al., 2024c) | 79.0 | 90.3 | 90.8 | 96.2 | 93.4 | 98.0 | 99.0 | 79.9 | 90.8 |
| w/ **Surgery** (Yang et al., 2024a) | 80.3 | 90.8 | 94.3 | 98.2 | 94.1 | 98.7 | 99.2 | 82.5 | 92.3 |
| w/ **ProbSurgery (Ours)** | 80.7 | 91.4 | 96.2 | 99.6 | 95.9 | 99.2 | 99.5 | 84.1 | 93.8 (1.5) |
| **SOTA**: EMR-MERGING (Huang et al., 2024) | 83.2 | 90.7 | 96.8 | 99.7 | 97.9 | 99.1 | 99.7 | 82.7 | 93.7 |

*Table 11.* Multi-task performance when merging ViT-B/16 models on eight tasks. The green value denotes the improvement of ProbSurgery compared with the counterpart method Surgery (Yang et al., 2024a).

| Method | SUN397 | Cars | RESISC45 | EuroSAT | SVHN | GTSRB | MNIST | DTD | Avg. |
|---|---|---|---|---|---|---|---|---|---|
| Pretrained | 63.8 | 64.6 | 65.7 | 54.5 | 52.0 | 43.3 | 51.7 | 45.1 | 55.0 |
| Individual | 81.8 | 86.8 | 96.9 | 99.7 | 97.8 | 99.1 | 99.7 | 82.0 | 92.9 |
| Fisher Merging (Matena & Raffel, 2022) | 68.5 | 69.9 | 75.2 | 80.4 | 73.2 | 61.2 | 94.5 | 50.7 | 71.7 |
| RegMean (Jin et al., 2023) | 69.1 | 71.6 | 77.6 | 88.8 | 83.7 | 70.2 | 96.9 | 54.6 | 76.6 |
| Weight Averaging | 67.7 | 70.0 | 75.3 | 79.5 | 74.9 | 60.1 | 94.4 | 43.8 | 70.7 |
| w/ **Surgery** (Yang et al., 2024a) | 70.3 | 72.4 | 88.8 | 97.6 | 82.0 | 83.1 | 98.1 | 68.5 | 82.6 |
| w/ **ProbSurgery (Ours)** | 74.0 | 79.4 | 95.7 | 99.7 | 87.8 | 98.8 | 99.4 | 81.2 | 89.5 (6.9) |
| Task Arithmetic (Ilharco et al., 2023) | 61.1 | 65.9 | 74.0 | 76.2 | 88.0 | 73.9 | 98.4 | 53.0 | 73.8 |
| w/ **Surgery** (Yang et al., 2024a) | 68.3 | 72.3 | 88.7 | 97.7 | 91.0 | 89.5 | 98.9 | 72.9 | 84.9 |
| w/ **ProbSurgery (Ours)** | 71.9 | 79.3 | 95.8 | 99.7 | 93.5 | 99.0 | 99.4 | 81.5 | 90.0 (5.1) |
| Ties-Merging (Yadav et al., 2023) | 69.1 | 72.5 | 80.5 | 84.0 | 85.0 | 71.5 | 98.1 | 54.9 | 77.0 |
| w/ **Surgery** (Yang et al., 2024a) | 73.0 | 76.2 | 90.7 | 98.1 | 89.7 | 86.7 | 98.7 | 75.2 | 86.0 |
| w/ **ProbSurgery (Ours)** | 75.1 | 80.9 | 95.8 | 99.7 | 92.2 | 99.0 | 99.5 | 82.0 | 90.5 (4.5) |
| LW AdaMerging (Yang et al., 2024c) | 70.2 | 80.7 | 81.6 | 94.8 | 91.6 | 95.8 | 98.5 | 66.2 | 84.9 |
| w/ **Surgery** (Yang et al., 2024a) | 73.6 | 81.5 | 90.4 | 98.5 | 93.2 | 97.4 | 98.9 | 77.0 | 88.8 |
| w/ **ProbSurgery (Ours)** | 75.3 | 83.7 | 95.5 | 99.8 | 94.9 | 99.0 | 99.4 | 82.3 | 91.2 (2.4) |

# D. Visualization Results

**Representation visualization.** In Figures 9, we visualize the generated representation via t-SNE technique, which clearly illustrates the advantages of our proposed method, ProbSurgery, in generating high-quality representations as a

post-calibration technique for model merging. Its advantages can be divided into three parts, including

- **Better Alignment** is evident as the red points produced by ProbSurgery closely align with the blue points representing individual task-specific models, preserving critical features and ensuring the merged representations stay faithful to the original tasks.

- **Reduced Overlap** is prominently shown across datasets such as DTD and SVHN, where ProbSurgery minimizes the interference between tasks by creating well-separated and distinct clusters, in contrast to the significant overlap seen in both four baseline methods like Weighted Averaging.

- **Task-Specific Robustness** is highlighted by ProbSurgery's ability to consistently produce clear and tightly clustered representations across diverse tasks, including both simpler datasets like MNIST and more complex ones like EuroSAT, demonstrating its adaptability and effectiveness in handling a wide range of scenarios.

These strengths establish ProbSurgery as a superior method for generating interpretable, task-preserving, and reliable representations.

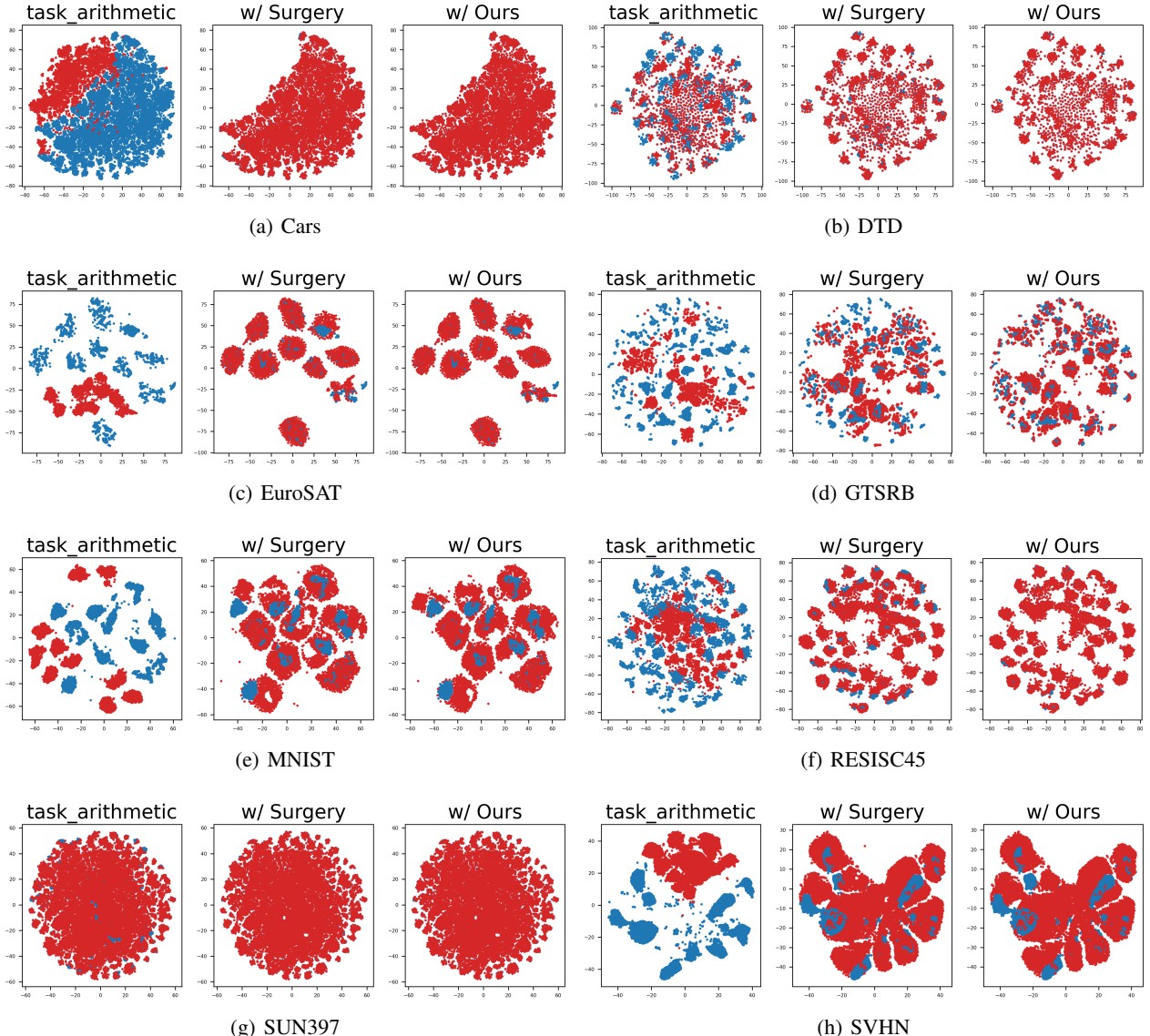

*Figure 9.* Comparisons of varying generated representations, where the baseline is **Task Arithmetic** and the backbone is **ViT-B/32**.

