# OpenReview forum: "Representation Surgery in Model Merging with Probabilistic Modeling"
_ICML.cc/2025/Conference — ICML 2025 poster_

### Official Review · Reviewer_HxL4 · 2025-03-09

**Overall Recommendation:** 4

**Summary:**

This paper builds upon an earlier work, representation survey. It there are T tasks and 1 model that is trained on each task, representation surgery paper defined representation bias as the sum of the distances between the merged model and the model trained on the current task for all the tasks. To alleviate this bias, they train an MLP module for each task that ensures that the representations of the merged model are closer to the model trained on the given task. This paper uses a gaussian distribution to model the latent space of the representation bias and uses an MLP module for each task to learn the mean and the standard deviation of the distribution, where the objective is to minimize the representation bias over all the tasks. They also formulate direct feature alignment (DFA) and proxy distribution alignment (PDA) methods to learn one module for all the tasks. They show that the  PDA consistently outperforms the surgery based merging on all the model merging tasks. They also demonstrate that the model merged using method generalizes on out of distribution and on domain drift when used in conjunction with weight averaging.

**Claims And Evidence:**

Claims:
1. To calibrate the representations of the outputs of the models merged using various methods and they demonstrated the same using their experiments. They used 4 model merging methods - Weight Averaging, Task Arithmetic, TIES, ADAMerging. The evidence is in table 2,3,5.
2. To use one unified module for all the tasks. The evidence is in table 4 and figure 8.
3. Tackling out of domain distribution and domain shift challenges. The evidence is in table 5.

**Essential References Not Discussed:**

All the essential references are discussed

**Experimental Designs Or Analyses:**

They did a lot of ablation studies such as the values of the hidden size h2, the sampling parameter, the various loss functions etc. They also showed that their method converges faster when compared to surgery and the efficacy of calibration by comparing the l1 distance between the representations yielded by the probsurgery and surgery methods when compared to the representation of the model trained on that task.

**Methods And Evaluation Criteria:**

The datasets used by this method have been used in the past by other model merging methods such as adamerging. In addition to the vision domain, they also evaluated their method on NLP tasks.

**Other Comments Or Suggestions:**

You could also explore correcting the representation bias of all the layers rather than the last layer along the lines of surgeryv2

TYPO:
In the caption of table4, both learning with eight modules and learning with one module are denoted by cross.

**Other Strengths And Weaknesses:**

Strengths:
1. The paper is very well written. The experiments section is very elaborate and bolsters all the claims
2. This method is parameter efficient and requires only 1 module when compared to the representation surgery method

Weakness:
1. There is a risk that the merged model might lose its general capabilities. It is important to also show that the merged model performs well on datasets such as Imagenet for classification, coco dataset for object detection etc.

**Questions For Authors:**

1. Your method is effective as the parameter interference hampers the model's performance, thus requiring this post-hoc calibration. Would this method still be able to improve methods such as DARE which try to reduce the parameter interference by reducing the number of parameters to begin with? Why is the performance improvement less for ada-merging while it is more pronounced for others? Similarly even for the NLP tasks, the gains are not as significant as the vision domain. Is it because of a smaller model?
2. The performance of model merging methods deteriorates as the number of models keep increasing. While your method alleviates it, would it still be effective if the number of models are 2 or 3?

**Relation To Broader Scientific Literature:**

This method corrects the representations of the model yielded by various model merging methods in a post-hoc manner to make the representation closer to that of the model trained on the current task. This work builds upon an earlier work called Representation surgery. They were able to show gains over the representation surgery method in multi-task learning, domain drift and out of distribution data settings. While the representation surgery method required one module for each task, this method can improve over it using only one module for all the tasks.

**Theoretical Claims:**

Ya, they claimed that the probabilistic method results in a smaller classification error than a deterministic method. They proved it using PAC-Bayes theorem

---

> ### Author Rebuttal · Authors · 2025-04-01
>
> We appreciate your insightful comments and give the response as follows.
>
> **Q1 Generalization on ImageNet**: Thanks for your valuable suggestion. We try the generalization performance of the merged model on unseen tasks like Classification task (ImageNet1k). We follow the setting in our paper, i.e., merging eight expert models, and then verfiy its general capability on the test set of ImageNet1k. The backbone is ViT-B/32 and the baseline is Task Arithmetic. The results in the table below show that the baseline Task Arithmetic method achieves 27.6% top-1 test accuracy, while the Surgery and ProbSurgery methods improve it to 48.5% and 52.7%, respectively. Thus, we believe our proposal represents a significant advancement in enhancing the generalization performance of existing model merging techniques.
>
> | Method | Test Acc |
> | - | - |
> |Task Arithmetic   | 27.6  |
> | w/ Surgery         | 48.5  |
> | w/ ProbSurgery | 52.7  |
>
>
> **Q2 ProbsurgeryV2, tntegrating Probsurgery into each block**: Thanks for this valuable suggestion. Our proposed ProbSurgery also can integrated into each block of the ViT-based model to surgery the layer (or block)-level representation bias, which is more refined and contributes to better performance. By adopting the design in Surgery V2, we conduct experiments on Task Arthmetric with ViT-B/32 and show the result in the following table.
>
> | Method | SUN397 | Cars | RESISC45 | EuroSAT | SVHN | GTSRB | MNIST | DTD | Avg |
> | - | - | - | - | - | - | - | - | - | - |
> | Task Arithmetic   | 55.2 | 54.9 | 66.7 | 78.9 | 80.2 | 69.7 | 97.3 | 50.4 | 69.1 |
> | w/ Surgery        | 63.8 | 59.9 | 83.3 | 97.9 | 87.0 | 87.0 | 98.6 | 69.4 | 80.9 |
> | w/ ProbSurgery    | 67.0 | 67.0 | 94.1 | 99.8 | 91.2 | 98.8 | 99.4 | 79.0 | 87.0 |
> | w/ Surgery v2     | 73.8 | 67.9 | 94.5 | 99.6 | 96.8 | 98.8 | 99.5 | 78.0 | 88.6 |
> | w/ ProbSurgery v2 | 74.1 | 68.2 | 94.8 | 99.8 | 97.1 | 99.1 | 99.8 | 78.3 | 88.9 |
>
>
> **Q3 Typos**: Thanks for pointing out these typos. We will correct these typos in the next version of our manuscript.
>
> **Q4-1 Other merging methods**. Our method is applicable to all model merging approaches, not just those based on task vectors. The table below reports the performance of three non-task-vector methods—Fisher Merging, RegMean, Task Arithmetic + DARE, and EMR-Merging — using ViT-B/32 as the backbone. We can see that applying ProbSurgery consistently improves average accuracy. In conclusion, we believe that ProbSurgery's effectiveness has been fully validated.
>
> | Method | SUN397 | Cars | RESISC45 | EuroSAT | SVHN | GTSRB | MNIST | DTD | Avg |
> | - | - | - | - | - | - | - | - | - | - |
> | Fisher Merging | 68.6 | 69.2 | 70.7 | 66.4 | 72.9 | 51.1 | 87.9 | 59.9 | 68.3 |
> | w/ ProbSurgery | 71.7 | 73.1 | 91.3 | 99.7 | 91.2 | 96.1 | 99.5 | 76.8 | 87.4 |
> | RegMean        | 65.3 | 63.5 | 75.6 | 78.6 | 78.1 | 67.4 | 93.7 | 52.0 | 71.8 |
> | w/ ProbSurgery | 71.5 | 72.7 | 94.1 | 99.0 | 93.0 | 97.7 | 99.3 | 78.3 | 88.2 |
> | Task Arithmetic + DARE|  51.7 | 51.4 | 63.2 | 75.4 | 76.7 | 66.2 | 94.9 | 46.9 | 65.8 |
> | w/ ProbSurgery |  67.1 | 65.1 | 93.2 | 98.8 | 90.3 | 97.8 | 98.4 | 78.1 | 86.1 |
> | EMR\-Merging   | 75.2 | 72.8 | 93.5 | 99.5 | 96.9 | 98.1 | 99.6 | 74.4 | 88.7 |
> | w/ ProbSurgery | 75.3 | 73.3 | 94.1 | 99.7 | 97.2 | 98.4 | 99.7 | 75.1 | 89.1 |
>
> **Q4-2 Improvement in Adamerging**: The improvement observed with Adamerging is not significant, as it is a more advanced merging method that results in representation bias during merging (as shown in Fig 4, a smaller distance is achieved by Adamerging). Therefore, when integrated with ProbSurgery, the improvement is not considerable.
>
> **Q4-3 NLP tasks**: Since different NLP tasks always enjoy the same parameters in almost all layers or blocks given a model like Bert, it exhibits fewer representation bias compared to vision tasks, which are highly sensitive to parameter changes. This difference leads to minimal performance degradation when merging multiple NLP tasks. Thus, integrating our proposal with NLP tasks yields a relatively smaller performance improvement than with vision tasks.
>
>
> **Q5 Merging fewer models**: We believe that any model fusion process, regardless of the number of models involved, induces parameter changes that lead to representation bias. Consequently, our method can alleviate this bias and enhance performance. Due to the word limit, we merged three models and present the results in the following table. Notably, our method also demonstrates performance gains even when merging a small number of models.
>
> | Method            | SUN397 | Cars | RESISC45 | Avg |
> | -                 | - | - | - | - |
> | Task Arithmetic   | 71.8 | 69.5 | 90.1 | 77.1 |
> | w/ ProbSurgery    | 74.1 | 73.8 | 94.4 | 80.8 |

---

> > ### Comment · Reviewer_HxL4 · 2025-04-08
> >
> > I thank the authors for answering all my questions. I would like to keep my score.

---

### Official Review · Reviewer_ACu4 · 2025-03-11

**Overall Recommendation:** 4

**Summary:**

This paper provides a probabilistic interpretation of the previous work, Surgery, which addresses the issue of representation bias. Additionally, two strategies are proposed to reduce overall training costs, supported by a theoretical analysis that highlights the advantages of the probabilistic approach. Extensive experiments demonstrate the superiority of the proposed method.

**Claims And Evidence:**

Yes, the paper addresses a valuable problem, and the motivation for the proposed method is clearly articulated. The algorithm is supported by the theoretical analysis.

**Essential References Not Discussed:**

This paper provides a comprehensive introduction to the task background and relevant preliminaries. There are no existing papers that are closely related to this work that require further discussion.

**Experimental Designs Or Analyses:**

The authors provide rational and rigorous experimental designs of main experiments and ablation studies. The analyses are comprehensive and insightful

**Methods And Evaluation Criteria:**

Yes, the paper introduces a probabilistic approach to model representation bias, leading to more robust estimations. The datasets used in the experiments are widely applicable. In addition to conventional metrics, the authors incorporate visualization techniques to illustrate biased representations.

**Other Comments Or Suggestions:**

None.

**Other Strengths And Weaknesses:**

Strengths:
1) The writing in this paper is exceptionally clear, with the author providing detailed background information and a well-articulated motivation for the study.

2) The proposed method appears highly reproducible. The probabilistic approach is both simple and effective, successfully modeling the uncertainty that implicitly arises from parameter interference when merging multiple models.

3) The theoretical analysis demonstrates that the classification error of the probabilistic approach is lower than that of a deterministic approach, enhancing the novelty of the proposed method.

Weaknesses and questions:
1) There is an enhanced version of Surgery, named SurgeryV2 [1], which incorporates the surgery module into each block of the encoder. I am curious to know whether the proposed ProbSurgery also implements this operation.

2) The proposed ProbSurgery shows potential for improved performance when integrated with existing model merging methods. I noticed that the selected baselines belong to weight-based approach. I have some concerns regarding the applicability of ProbSurgery to other types of model merging methods. For instance, could it be integrated with route-based methods like EMR-Merging [2]?

3) Impact of the scale of validation set. The proposed framework relies on unlabeled test/validation data to build self-supervision signals and update the parameters of the ProbSurgery module. It would be beneficial for the author to discuss the impact of varying amounts of available test data on model performance.

4) Recent methods [2][3] have begun to evaluate their performance under extreme conditions, such as model merging with tens of models. I suggest the author conduct experiments under this setting to validate the generalization and robustness of ProbSurgery.

[1] SurgeryV2: Bridging the Gap Between Model Merging and Multi-Task Learning with Deep Representation Surgery. ArXiv 2024
[2] Emr-Merging: Tuning-free high-performance model merging. NeurIPS 2024
[3] Localizing Task Information for Improved Model Merging and Compression. ICML 2024

**Questions For Authors:**

See Weaknesses.

**Relation To Broader Scientific Literature:**

This paper proposes a novel approach to bridge the gap between multiple individual expert models and the merged model through a representation rectification operation. Experimental results demonstrate that the proposed method, ProbSurgery, achieves performance comparable to that of various expert models. Consequently, I believe this approach has the potential to enhance the applicability of model merging as a multi-task learning strategy.

**Theoretical Claims:**

Yes, I checked all contents for the theoretical claims and didn't find significant errors.

---

> ### Author Rebuttal · Authors · 2025-04-01
>
> We appreciate your insightful comments and give the response as follows.
>
> **Q1 ProbsurgeryV2, integrating Probsurgery into each block**: In this paper, ProbSurgery is integrated into the last layer. However, it also can be integrated into each block of the ViT-based model to surgery the layer (or block)-level representation bias, which is more refined and contributes to better performance. By adopting the design in Surgery V2, we conduct experiments on Task Arthmetric with ViT-B/32 and show the result in the following table. When our method is applied after every block, it can more effectively reduce representation bias and achieve better performance.
>
> | Method | SUN397 | Cars | RESISC45 | EuroSAT | SVHN | GTSRB | MNIST | DTD | Avg |
> | - | - | - | - | - | - | - | - | - | - |
> | Task Arithmetic   | 55.2 | 54.9 | 66.7 | 78.9 | 80.2 | 69.7 | 97.3 | 50.4 | 69.1 |
> | w/ Surgery        | 63.8 | 59.9 | 83.3 | 97.9 | 87.0 | 87.0 | 98.6 | 69.4 | 80.9 |
> | w/ ProbSurgery    | 67.0 | 67.0 | 94.1 | 99.8 | 91.2 | 98.8 | 99.4 | 79.0 | 87.0 |
> | w/ Surgery v2     | 73.8 | 67.9 | 94.5 | 99.6 | 96.8 | 98.8 | 99.5 | 78.0 | 88.6 |
> | w/ ProbSurgery v2 | 74.1 | 68.2 | 94.8 | 99.8 | 97.1 | 99.1 | 99.8 | 78.3 | 88.9 |
>
>
> **Q2 Integrating with other methods**: Due to page restriction, we keep the same setting with Surgery that only conducts experiments on four baselines based on task vectors. Our method ProbSurgery is completely orthogonal to existing model merging approaches and can be incorporated into any model merging method to solve its representation bias problem. In the following table, we integrate ProbSurgery with other three methods that are not based on task vectors, including Fisher Merging, Regmean, and EMR\-Merging. By integrating our method, each baseline sees a substantial boost in performance, highlighting the effectiveness of our approach in mitigating representation bias.
>
> | Method | SUN397 | Cars | RESISC45 | EuroSAT | SVHN | GTSRB | MNIST | DTD | Avg |
> | - | - | - | - | - | - | - | - | - | - |
> | Fisher Merging | 68.6 | 69.2 | 70.7 | 66.4 | 72.9 | 51.1 | 87.9 | 59.9 | 68.3 |
> | w/ ProbSurgery | 71.7 | 73.1 | 91.3 | 99.7 | 91.2 | 96.1 | 99.5 | 76.8 | 87.4 |
> | RegMean        | 65.3 | 63.5 | 75.6 | 78.6 | 78.1 | 67.4 | 93.7 | 52.0 | 71.8 |
> | w/ ProbSurgery | 71.5 | 72.7 | 94.1 | 99.0 | 93.0 | 97.7 | 99.3 | 78.3 | 88.2 |
> | EMR\-Merging   | 75.2 | 72.8 | 93.5 | 99.5 | 96.9 | 98.1 | 99.6 | 74.4 | 88.7 |
> | w/ ProbSurgery | 75.3 | 73.3 | 94.1 | 99.7 | 97.2 | 98.4 | 99.7 | 75.1 | 89.1 |
>
> **Q3 The size of the validation set**
> Thanks for this constructive suggestion. We made an experiment to verify how the size of the validation set impacts ProbSurgery's performance. In the following table, we can see that using a larger proportion of the validation set helps the ProbSurgery module better capture and mitigate inherent biases, leading to higher overall accuracy. Notably, even when only 10% of the unlabeled validation data is used, our method still exceeds Surgery’s performance of 80.9%, demonstrating its robustness under limited data conditions.
>
>
> | Method | Ratio of Val set | SUN397 | Cars | RESISC45 | EuroSAT | SVHN | GTSRB | MNIST | DTD | Avg |
> | - | - | - | - | - | - | - | - | - | - | - |
> | Task Arithmetic | -  | 55.2 | 54.9 | 66.7 | 78.9 | 80.2 | 69.7 | 97.3 | 50.4 | 69.1 |
> | w/ Ours | 10% | 63.5 | 64.5 | 89.6 | 94.9 | 88.4 | 89.4 | 98.4 | 74.5 | 82.9 |
> | w/ Ours | 50% | 65.9 | 66.5 | 92.3 | 97.8 | 90.5 | 95.0 | 98.8 | 77.2 | 85.5 |
> | w/ Ours | 100%| 67.0 | 67.0 | 94.1 | 99.8 | 91.2 | 98.8 | 99.4 | 79.0 | 87.0 |
>
>
> **Q4 Merging tens of models**: Thanks for this constructive suggestion. To verify the performance in more challenging settings, we additionally add three tasks (ImageNet100, CIFAR100, and real-world Homeoffice) to original eight tasks. The results in the following table show that even when merging a large number of models, our method can effectively correct the representation bias and achieve superior performance.
>
>
> | Method | SUN397 | Cars | RESISC45 | EuroSAT | SVHN | GTSRB | MNIST | DTD | ImageNet100 | CIFAR100 | HomeOffice | Avg |
> | - | - | - | - | - | - | - | - | - | - | - | -|-|
> | Task Arithmetic  |31.8 | 39.3 | 49.6 | 60.3 | 74.4 | 60.4 | 95.8 | 38.4 | 11.6 | 42.6 | 70.3 | 52.3 |
> | w/ Surgery | 50.5 | 52.7 | 76.8 | 96.3 | 83.8 | 79.9 | 98.6 | 65.8 | 56.8 | 55.6 | 80.3 | 72.5 |
> | w/ ProbSurgery  | 53.7 | 55.7 | 86.3 | 97.9 | 85.5 | 96.0 | 98.8 | 72.9 | 74.8 | 63.4 | 86.2 | 79.2 |

---

> > ### Comment · Reviewer_ACu4 · 2025-04-03
> >
> > The rebuttal has solved my question, and I will keep the score.

---

### Official Review · Reviewer_teo3 · 2025-03-13

**Overall Recommendation:** 3

**Summary:**

Surgery is a method to improve the merging performance by reducing the representation bias of model merging. This paper argues that there are two main issues of Surgery. First, the representation discrepancy is not fully addressed. Second, Surgery requires multiple task-specific modules rather than a unified one. This paper proposes ProbSurgery to mitigate these. They adapt the latent variable model (e.g., VAE) to enhance Surgery and further extend it to the one-to-all setting. Empirical and theoretical results are shown to illustrate their method's benefit.

# After rebuttal:

My concerns are addressed and I will raise my score.

**Claims And Evidence:**

Some illustration figures and motivations are unclear to me. Please see the questions.

**Essential References Not Discussed:**

N/A

**Experimental Designs Or Analyses:**

I have no questions.

**Methods And Evaluation Criteria:**

It may lack some important baselines. The motivation and design of the method are not clear to me. Please see the questions.

**Other Comments Or Suggestions:**

Typos:
- Title of Sec.3.1: represetative - >  representative
- Def.3.2: $f_{\theta_T}$ - >  $\{f_{\theta_t}\}_{t=1}^T$

**Other Strengths And Weaknesses:**

The novelty and originality of this paper are good as they address two main issues of the previous method by applying VAE and Surgery. However, I think the clarity could be improved. Also, some important baselines are missing in their experiments, which may limit the significance of their paper. Please see the questions for details.

**Questions For Authors:**

1. How to compute $G$ in Observation.3.3.? I think a detailed formulation would be better than a description “overall performance gap”.
2. What is the upper performance in Fig.2(a)?
3. Why is the ablation of (Prob)Surgery not studied with Fisher and RegMean merging in Tab.2?
4. In Eq.4, why do we align the shift of merged features $z_{i,t}^{\theta_{unif}}-\xi_{i,t}$ rather than $z_{i,t}^{\theta_{unif}}$ with $z^{\theta_t}$? I understand that Eq.4 is an extension of Eq.1, but I am not clear why we follow the objective in Eq.1.
5. I do not understand the following sentence in lines 217- 219: "Due to limited ... to other tasks". Is the limited representation capability the same as what is illustrated in Fig.1? What do "other tasks" mean?
6. I am curious about which theorem in (McAllester, 2003) the author adapted in their proposed Theorem.4.1.
7. What is $E[Q_{\omega}, z^{bias}]$ in Eq.9?
8. As the proposed method needs to retrain the (Prob)Surgery modules after merging, I think it would be fair to compare it with the performance of the merged model after training on all validation sets.

---

My concerns are addressed and I will raise my score.

**Relation To Broader Scientific Literature:**

Model merging is an emergent technique but suffers from the performance gap between merged and task-specific models. This paper's method is built on a previously proposed method to address two main issues. I believe the proposed method can reduce both the memory cost and inference time of the merged model, as well as improve the merging performance compared to the previous one. Moreover, as a post-merging method, this paper is compatible with other merging methods, which enhances the impact of this paper.

**Theoretical Claims:**

I wonder about the adaptation of Theorem.4.1 from literature. See the questions for details.

---

> ### Author Rebuttal · Authors · 2025-04-01
>
> We appreciate your insightful comments and give the response as follows.
>
> **Q1 The observation**: Here, $G(\cdot)$ quantifies the difference in test accuracy, which reflects the average discrepancy between the feature representations produced by these two models. For a more formal representation, we extend the function $G(\cdot)$ to:
> $E_{t\sim[T]} \Big[\text{Acc}(f_{\theta_t}, D_t) - \text{Acc}(f_{\theta_{unif}}, D_t)\Big]
> \propto E_{t\sim[T]} \xi_t,$ where $\text{Acc}(f, D)$ denotes the test accuracy of model $f$ on the test set $D$. Detailed explanations will be added to our manuscript in subsequent revisions.
>
> **Q2 Upper performance in Fig2**: The upper performance in Figure 2 denotes the average performance of various expert models on corresponding merged tasks.
>
> **Q3 More baselines**: Due to page restrictions, we keep the same setting with Surgery that makes experiments on four methods based on task vectors. (Prob)Surgery is completely orthogonal to existing model merging methods and can be incorporated into all of them to mitigate the representation bias. In the table below (*more detailed results can be found in the response to Reviewer HxL4, Q4-1*), it shows that integrating (Prob)Surgery into existing baselines significantly improves performance. This confirms that (Prob)Surgery effectively mitigates representation bias in model merging and can be seamlessly incorporated into any merging approach.
> | Method  | Avg |
> | -  | - |
> | Fisher Merging | 68.3 |
> | w/ Surgery  | 82.6 |
> | w/ ProbSurgery | 87.4 |
> | RegMean | 71.8 |
> | w/ Surgery | 82.9 |
> | w/ ProbSurgery | 88.2 |
>
> **Q4 Format of Eq. (4)**: We follow the objective in Eq. (1) to maintain the core motivation of Surgery, namely to remove the representation bias introduced by the merged model. Concretely, $z_{i,t}^{\mathrm{unif}} - \xi_{i,t}$ represents the corrected (post‐calibrated) representation rather than the raw merged representation $z_{i,t}^{\mathrm{unif}}$. Extending from a deterministic correction term $\xi_{i,t}$ in Eq. (1) to a probabilistic treatment in Eq. (4) lets us more accurately capture the uncertainty arising from parameter interference during merging.
>
> **Q5 Statement in Lines 217-219**: The “limited representational capacity” refers to the deterministic nature of the Surgery module. While deterministic modeling can be simpler and faster to train, it struggles to capture the inherent uncertainty of multiple merged tasks. As Figure 1 shows, even after applying a deterministic correction, the merged features remain partially misaligned with each individual model’s distribution, implying that a single deterministic module does not generalize well beyond its initially learned task. Besides, “other tasks” denote the remaining tasks in the merged set, which the single deterministic Surgery module fails to accommodate.
>
> **Q6 Theorem 4.1**: In McAllester (2003), the statement closest to the paper’s Theorem 4.1 is Theorem 1, often referred to as the “PAC-Bayesian Theorem”. It provides a general PAC-Bayes bound on the expected loss of a posterior hypothesis in terms of its empirical loss plus a KL‐divergence term from the prior—precisely the structure adapted in ours.
>
> **Q7 Expectation in Eq. (9)**: Eq. (9) measures how a predicted distribution $Q_{\omega}$ fits the observed (true) distribution $P^{\mathrm{bias}}$. We take the expectation of that score with respect to the “true” distribution $P^{\mathrm{bias}}$: $S\bigl(Q_{\omega}, P^{\mathrm{bias}}\bigr) =\int S\bigl(Q_{\omega}, z^{\mathrm{bias}}\bigr)\mathrm{d}P^{\mathrm{bias}}(z^{\mathrm{bias}}). $ Finally, it is written as $\mathop{\arg \min}\limits_{\omega} \mathcal{S}(Q_\omega, P^{\rm bias}) := \mathop{\arg \min}\limits_{\omega} E_{z^{\rm bias} \sim P^{\rm bias}}[Q_\omega, z^{\rm bias}]$, as shown in Eq. (9). Due to only one observation $z^{\rm bias}$ (a fixed bias) in practice, this equation can omit the expectation form and be reformulated as $\mathop{\arg \min}\limits_{\omega} \mathcal{S}(Q_\omega, P^{\rm bias}) := \mathop{\arg \min}\limits_{\omega}  \mathcal{S}(Q_\omega, z^{\rm bias})$. Thus, in Eq. (10), we only have one variable $z^{\rm bias}$ and estimate the distribution $Q_\omega$ via sampling.
>
> **Q8 Validation set**: First, the post-training step in (Prob)Surgery relies on an extra validation set that does not require any labels. Acquiring such a small-scale set is feasible in the real world since no label annotations are required. Meanwhile, to ensure a fair comparison, we employ test-time adaptation (TTA, adopted in many model merging methods like Adamerging) to post-train a merged model with Task Arithmetic. The results in the table below clearly demonstrate that our method significantly outperforms the TTA approach.
> | Ratio of val set | 10% | 50% | 100% |
> | - | - | - | - |
> | Task Arithmetic: 69.1 |
> |w/ TTA | 72.8 | 74.1 | 75.4 |
> |w/ Ours | 84.8 | 86.2 | 87.0 |
>
> **Q9 Typos**: Thanks for pointing out these typos. We will correct these in the next manuscript.

---

> > ### Comment · Reviewer_teo3 · 2025-04-05
> >
> > Thanks a lot for your response. However, I have a new question here. When merging the models, how do the authors process the classifiers? You maintain task-specific classifiers and use them according to the task index, or have other methods? If it is the former, how do we do OOD evaluation?

---

> > > ### Author Response · Authors · 2025-04-06
> > >
> > > Thank you for your response. Here are the details of our OOD detection experiments:.
> > >
> > > When we finish the model merging and the post-calibration process with (Prob)Surgery, we test the performance of different model merging methods on OOD detection. We use the test sets of four tasks and a small OOD set of 500 samples randomly sampled from the test set of SUN397, denoted by $D_1,D_2, D_3, D_4$, and $D_o$.
> > >
> > > 1. Test Accuracy
> > >    To measure test accuracy, we follow the standard model-merging approach and maintain separate task-specific classifiers for the merged tasks. Formally, the test accuracy on the test set of the $t$-th task $D_t$ can be expressed as: $ACC_{t} = \frac{\sum_{(x,y) \in {D_t + D_o}} 1({\hat y} = y)}{||D_t + D_o||}$, where ${\hat y}$ denotes the predicted label.
> > >
> > > 2. AUROC
> > >    To report AUROC, we use a commonly adopted unsupervised technique that does not rely on the classifier. Specifically, for each task, we store its class prototypes during training (when no OOD samples are present). In the test phase, for task $t$, we compute the representation of every sample in the test set $D_t+D_o$ and measure its distance to the stored class prototypes. The greater the distance, the more likely the sample is to be an outlier. Eventually, the AUROC value can be calculated on the distance and the 0-1 label (where 0 and 1 denote the In-distribution and Out-of-distribution samples, respectively).
> > >
> > > These details will be added into the next version of our manuscript.

---

### Official Review · Reviewer_XTHh · 2025-03-13

**Overall Recommendation:** 4

**Summary:**

This paper proposes ProbSurgery, a probabilistic approach to post-merging representation correction in model merging. The authors address the representation bias that occurs when merging multiple models for multitask learning. Unlike prior deterministic approaches (e.g., Surgery), ProbSurgery models the bias as a probabilistic distribution, which better handles uncertainty in parameter merging. The paper provides theoretical insights using a PAC-Bayes framework and proposes a one-to-all extension of ProbSurgery, allowing for efficient multi-task calibration. Experimental results across vision and NLP tasks show that ProbSurgery consistently outperforms existing merging strategies, achieving better generalization and robustness, particularly in OOD and domain shift settings.

**Claims And Evidence:**

Yes, all claims are supported by both theoretical and experimental evidence. 1) The main claim that ProbSurgery improves representation correction and generalization is well-supported by both theoretical and experimental evidence. 2) The paper uses a PAC-Bayes theoretical framework to show that modeling bias as a distribution leads to a lower classification error.

**Essential References Not Discussed:**

The cited works cover model merging and uncertainty modeling, making the context of the study clear.

**Experimental Designs Or Analyses:**

Yes, the experimental designs are sound
1) The experiments use strong baselines, including Weighted Averaging, Task Arithmetic, and AdaMerging, ensuring fair comparisons.
2) The inclusion of OOD and domain shift experiments strengthens the claim that ProbSurgery improves generalization.

**Methods And Evaluation Criteria:**

Yes, the utilized criteria are reasonable. 1) The problem of representation bias in model merging is well-motivated, and the paper systematically compares ProbSurgery to existing merging methods (e.g., Weighted Averaging, Task Arithmetic, AdaMerging). 2) The use of L1 distance and test accuracy as evaluation metrics is reasonable and aligns with prior work (Figures 2, 4).

**Other Comments Or Suggestions:**

This paper can be improved from the following three parts:
1) Discuss computational cost of ProbSurgery vs. deterministic methods.
2) Provide ablation studies on KL regularization λ and sampling variance.
3) Verify Gaussian assumptions for features.

**Other Strengths And Weaknesses:**

Strengths:
1) Novelty: Probabilistic modeling of representation bias is a significant advancement over deterministic approaches.
2) Theoretical Rigor: Strong PAC-Bayes justification for ProbSurgery's effectiveness.
3) Empirical Validation: Extensive experiments across vision and NLP tasks show clear improvements.
4) Real-world Relevance: The one-to-all setting makes ProbSurgery practical for multitask learning.

Weaknesses:
1) Computational Complexity: The additional probabilistic sampling step may introduce latency, but this is not discussed.
2) Sensitivity to Hyperparameters: The impact of hyperparameters like KL regularization $\lambda$ is not fully analyzed.
3) Gaussian Assumption: In this paper, the authors regularize the corrected representation as a normal Gaussian distribution. However, they do not provide sufficient motivation for this choice. The assumption that representation bias follows a Gaussian distribution requires further justification and discussion.

**Questions For Authors:**

See Weaknesses

**Relation To Broader Scientific Literature:**

Yes, the contribution and difference compared with previous works have been widely discussed. The paper builds on previous model merging works (Matena & Raffel, 2022; Ilharco et al., 2023) and extends the Surgery framework (Yang et al., 2024a). It also relates ProbSurgery to uncertainty modeling in probabilistic embeddings (Vilnis & McCallum, 2014).

**Theoretical Claims:**

Yes, I checked all theoretical claims and didn't find errors. The PAC-Bayes analysis is correctly applied and suggests that a probabilistic bias correction method results in a lower classification error bound.

---

> ### Author Rebuttal · Authors · 2025-04-01
>
> We appreciate your insightful comments and give the response as follows.
>
> **Q1 Computational Complexity**: In practical implementation, we only sample once to generate the representation bias in the ProbSurgery module, which does not incur any additional cost compared to the deterministic method. Our results demonstrate that a single sample is sufficient to achieve state-of-the-art performance—outperforming the previous Surgery method. In the next version of our manuscript, we will provide further insights and discussions on how the number of samples impacts the algorithm's performance.
>
> Besides, we agree with your viewpoint that sampling multiple times can probably improve the estimation accuracy of the representation bias, thereby enhancing overall model performance. Specially, we sample $\rho$ times and compute their average values during model training. However, even if we perform multiple samplings, it does not incur extra computational cost since this averaging operation only involves first-order derivatives during backpropagation. In the following table, we try different sampling numbers and compare their performance. We can observe that increasing the sampling number does not lead to a significant improvement in overall performance. Therefore, to ensure better scalability, we have chosen not to introduce this hyperparameter explicitly, and instead, we set it to 1 by default.
>
> |Sampling numbers|1|3|5|10|
> | - | - | - | - | - |
> |Average acc on eight tasks|86.74|86.87|86.90|86.91|
>
> **Q2 Sensitivity to Hyperparameters**: Based on our empirical experience, we have set the trade-off coefficient for the KL term to $1 \times 10^{-3}$. To further explore the algorithm's sensitivity to this hyperparameter, we conducted experiments with Weight Average on ViT-B/32 and show the result in the following table, which shows that the algorithm's average accuracy remains nearly constant across various hyperparameter values, demonstrating that our method is insensitive to this parameter.
>
> |the trade-off coefficient $\lambda$|$1\times10^{-4}$| $1 \times 10^{-3}$ | $1 \times 10^{-2}$ | $1 \times 10^{-1}$ |
> |-|-|-|-|-|
> |Average acc on eight tasks|86.17|86.70|86.40|86.02|
>
> **Q3 Gaussian Assumption for representation**: In this work, we treat the representation bias as a Gaussian mainly for tractability and flexibility. The normal distribution provides a compact parameterization ($\mu$ and $\sigma$) that is easy to optimize, while also serves as a high-entropy “baseline” choice—minimizing assumptions about the true shape of the bias. Although the real distribution may deviate from exact Gaussianity, empirical tests show that this assumption reliably captures the dominant uncertainty in merged representations.

---

### Decision · Program_Chairs · 2025-05-01

**Decision:**

Accept (poster)

**Comment:**

This paper introduces ProbSurgery, a probabilistic approach to post-merging representation correction in model merging. The method addresses representation bias—a key limitation in merging multiple task-specific models—by modeling bias as a Gaussian distribution rather than using deterministic corrections (as in prior work like Surgery).

ProbSurgery makes a significant methodological advance in model merging, combining theoretical soundness (PAC-Bayes) with practical impact (scalability, OOD robustness). The rebuttal convincingly addressed critiques, and new experiments (e.g., ProbSurgeryV2, 11-task merging) strengthened the contribution.